# Score-Based Multimodal Autoencoder

**Daniel Wesego**                                                                 *dweseg2@uic.edu*
*Department of Computer Science*
*University of Illinois Chicago*

**Pedram Rooshenas**                                                              *pedram@uic.edu*
*Department of Computer Science*
*University of Illinois Chicago*

**Reviewed on OpenReview:** *https://openreview.net/forum?id=JbuP6UV3Fk*

## Abstract

Multimodal Variational Autoencoders (VAEs) represent a promising group of generative models that facilitate the construction of a tractable posterior within the latent space given multiple modalities. Previous studies have shown that as the number of modalities increases, the generative quality of each modality declines. In this study, we explore an alternative approach to enhance the generative performance of multimodal VAEs by jointly modeling the latent space of independently trained unimodal VAEs using score-based models (SBMs). The role of the SBM is to enforce multimodal coherence by learning the correlation among the latent variables. Consequently, our model combines a better generative quality of unimodal VAEs with coherent integration across different modalities using the latent score-based model. In addition, our approach provides the best unconditional coherence. The code can be found at https://github.com/rooshenasgroup/sbmae

## 1 Introduction

The real-world data often has multiple modalities such as image, text, and audio, which makes learning from multiple modalities an important task. Multimodal VAEs are a class of multimodal generative models that are able to generate multiple modalities jointly. Learning from multimodal data is inherently more challenging than from unimodal data, as it involves processing multiple data modalities with distinct characteristics.

In order to learn the joint representation of these modalities, previous approaches generally preferred to encode them to latent distribution that governs the data distribution across different modalities. In general, we expect the following properties from a multimodal generative model:

**Multiway conditional generation:** Given the presence of certain modalities, it should be feasible to generate the absent modalities based on the existing ones (Shi et al., 2019; Wu & Goodman, 2018). The conditioning process should not be limited to specific modalities; rather, any modality should serve as a basis for generating any other modality.

**Unconditional generation:** If we have no modality present to condition on, we should be able to sample from the joint distribution so that the generated modalities are coherent (Shi et al., 2019). Coherence in this case means that the generated modalities represent the same concept that is expressed in the different modalities we have.

**Conditional modality gain:** When additional information is provided to the model through the observation of more modalities, there should be an enhancement in the performance of the generated absent modalities. In other words, the performance should consistently improve as the number of observed (given) modalities increases.

**Scalability:** The model should scale as the number of total modalities increases. Moreover, the model complexity shouldn't become computationally inefficient when adding more modalities (Sutter et al., 2020).

Other properties like joint representation where we want a representation that takes into account the statistics and properties of all the modalities (Srivastava & Salakhutdinov, 2014; Suzuki et al., 2017) can be easily learned from a multimodal model obtaining the above properties. Another property is weak supervision where we make use of multimodal data that isn't paired together (Wu & Goodman, 2018).

Naively using multimodal VAEs by training all combinations of modalities becomes easily intractable as the number of models to be trained increases exponentially. We will need to train $2^M - 1$ different combinations of models for each subset of the modalities. Since this is not a scalable approach, previous works have proposed different methods of avoiding this by constructing a joint posterior of all the modalities. They achieve this by modeling the joint posterior over the latent space $\mathbf{z}$: $q(\mathbf{z}|\mathbf{x}_{1:M})$, where $\mathbf{x}_{1:M}$ is the set of modalities. To ensure the tractability of the inference network $q$, prior works have proposed using a product of experts (Wu & Goodman, 2018), a mixture of experts (Shi et al., 2019), or in the generalized form, a mixture of the product of experts (MoPoE) (Sutter et al., 2021); among others.

After selecting how to fuse the posteriors of different modalities, these approaches then construct the joint multimodal ELBO and use it to train multimodal data. At inference time, they use the modalities that are observed to generate the missing modalities. Wu & Goodman (2018) intentionally adds ELBO subsampling during training to increase the model's performance on generating missing modalities at inference time. Mixture of experts and mixture of products of experts subsample modalities as part of their training process because the posterior model is a mixture model. Subsampling of the modalities, as pointed out by Daunhawer et al. (2022), results in a generative discrepancy among modalities. We also observe that conditioning on more modalities doesn't improve the performance of the generated modality on some multimodal VAEs which goes contrary to one's expectation. As a model receives additional information, it should perform better or should show no decrease in performance. Daunhawer et al. (2022) further concludes that these models cannot be useful for real-world applications at this point due to these failures.

To overcome these issues, instead of constructing a joint posterior, we try to explicitly model the joint latent space of individual VAEs: $p_\theta(\mathbf{z}_{1:M})$. The joint latent model learns the correlation among the individual latent space in a separate score-based model trained independently without constructing a joint posterior as the previous multimodal VAEs. The joint model can ensure prediction coherence by sampling from the score model while also maintaining the generative quality that is close to a unimodal VAE. And by doing so, we avoid the need to construct a joint multimodal ELBO. We use the independently trained unimodal VAEs to generate latent samples, and use those samples to train a score network that models the joint latent space. This approach is scalable as it only uses $M$ unimodal VAEs and one score model. Unconditional generation from the joint distribution can also be done by sampling from the score model respecting the joint distribution. Conditional generation is also possible by fixing the observed modalities and generating the rest. It also avoids the subsampling problems discussed by Daunhawer et al. (2022) as it doesn't need to train a joint multimodal ELBO. In addition to that, it provides the best unconditional results compared with the baselines while at the same time having a competitive conditional performance. Figure 1 describes the overall architecture of the model.

Our contributions include 1) we propose a novel generative multimodal autoencoder approach that satisfies most of the appealing properties of multimodal VAEs, supported using extensive experimental studies. 2) We introduce coherence guidance to improve the coherence of observed and unobserved modalities. 3) Our experimental result demonstrates that the proposed model performs well unconditionally and conditionally and, in addition, exhibits enhanced robustness against adversarial perturbations compared to baseline methods.

## 2 Methodology

Assuming the each data point consists of $M$ modalities: $\mathbf{x}_{1:M} = (\mathbf{x}_1, \mathbf{x}_2, \cdots, \mathbf{x}_M)$, our latent variable model describes the data distribution as $p(\mathbf{x}_{1:M}) = \sum_{\mathbf{z}_{1:M}} p(\mathbf{x}_{1:M}|\mathbf{z}_{1:M})p(\mathbf{z}_{1:M})$, where $\mathbf{z}_{1:M} = (\mathbf{z}_1, \mathbf{z}_2, \cdots, \mathbf{z}_M)$ and $\mathbf{z}_k$ is the latent vector corresponding to the $k$th modality. In contrast to common multimodal VAE setups

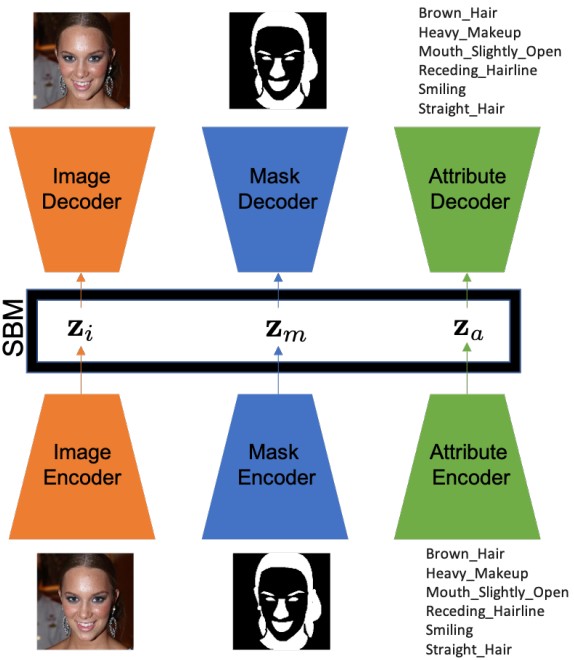

Figure 1: A variational or regularized auto-encoder will be used for each individual modality to get the latent representation which then will be used to train the score-based model which will allow the prediction of any modality given some or none. The auto-encoders are trained independently in the first stage and the respective $z$ of each modality will be used to train the score network.

(Wu & Goodman, 2018; Shi et al., 2019), we don't consider any shared latent representation among different modalities, and we assume each latent variable[1] $\mathbf{z}_k$ only captures the modality-specific representation of the corresponding modality $k$. Therefore, the variational lower bound on $\log p(\mathbf{x}_{1:M})$ can be written as:

$$\log p(\mathbf{x}_{1:M}) \geq \mathbb{E}_{q(\mathbf{z}_{1:M}|\mathbf{x}_{1:M})} \log \frac{p(\mathbf{x}_{1:M}|\mathbf{z}_{1:M})p(\mathbf{z}_{1:M})}{q(\mathbf{z}_{1:M}|\mathbf{x}_{1:M})}. \tag{1}$$

To simplify the joint generative model $p(\mathbf{x}_{1:M}|\mathbf{z}_{1:M})$ and the joint recognition model $q(\mathbf{z}_{1:M}|\mathbf{x}_{1:M})$, we assume two conditional indenpendencies: 1) Given an observed modality $\mathbf{x}_k$, its corresponding latent variable $\mathbf{z}_k$ is independent of other latent variables, i.e., $\mathbf{x}_k$ have enough information to describe $\mathbf{z}_k$: $\mathbf{z}_k \perp \mathbf{z}_{-k}|\mathbf{x}_k$. 2) Knowing the latent variable $\mathbf{z}_k$ of $k$th modality is enough to reconstruct that modality: $\mathbf{x}_k \perp \mathbf{x}_{-k}, \mathbf{z}_{-k}|\mathbf{z}_k$. Using these conditional independencies the generative and recognition models factorize as:

$$p(\mathbf{x}_{1:M}|\mathbf{z}_{1:M}) = \prod_{k=1}^{M} p(\mathbf{x}_k|\mathbf{z}_k) \text{ and } q(\mathbf{z}_{1:M}|\mathbf{x}_{1:M}) = \prod_{k=1}^{M} q(\mathbf{z}_k|\mathbf{x}_k). \tag{2}$$

Therefore we can rewrite the variational lower bound as:

$$\log p(\mathbf{x}_{1:M}) \geq \sum_{k} \mathbb{E}_{q_{\phi_k}(\mathbf{z}_k|\mathbf{x}_k)} \log p_{\psi_k}(\mathbf{x}_k|\mathbf{z}_k) - D_{\mathrm{KL}}(\prod_{k} q_{\phi_k}(\mathbf{z}_k|\mathbf{x}_k)||p_\theta(\mathbf{z}_{1:M})), \tag{3}$$

where $\phi_k$ and $\psi_k$ are the parameterizations of the recognition and generative models, respectively, and $\theta$ parameterizes the prior. If we assume the prior $p_\theta(\mathbf{z}_{1:M})$ factorizes as $\prod_k p_{\theta_k}(\mathbf{z}_k)$, then the variational lower bound of $\log p(\mathbf{x}_{1:M})$ becomes $\sum_k \mathrm{ELBO}_k$, where $\mathrm{ELBO}_k$ is the variational lower bound of individual modality. However, such an assumption ignores the dependencies among latent variables and results in a lack

---

[1]For simplicity we use "variable" to refer to the group of variables that describe the latent representation of a modality

of coherence among generated modalities when using prior for generating multimodal samples. To benefit from the decomposable ELBO but also benefit from the joint prior, we separate the training into two steps. In step I, we maximize the ELBO with respect to $\phi$ and $\psi$ assuming prior $p(\mathbf{z}_{1:M}) = \prod \mathcal{N}(\mathbf{0}, \sigma\mathbf{I})$, which only regularizes the recognition models. In step II, we optimize ELBO with respect to $\theta$ assuming a joint prior over latent variables. Therefore, in step II maximizing the ELBO reduces to: $\min_\theta D_{\text{KL}}(\prod_k q_{\phi_k}(\mathbf{z}_k|\mathbf{x}_k)||p_\theta(\mathbf{z}_{1:M}))$ and since recognition model is constant w.r.t. $\theta$, the step II becomes $\max_\theta \mathbb{E}_{p(\mathbf{x}_{1:M})}\mathbb{E}_{\prod_k q(\mathbf{z}_k|\mathbf{x}_k)} \log p_\theta(\mathbf{z}_{1:M})$, which is equivalent to maximum likelihood training of the parametric prior using sampled latent variables for each data points. And during inference, we only need samples from $p_\theta(\mathbf{z}_{1:M})$, thus we can parameterize $s_\theta(\mathbf{z}_{1:M}) \approx \nabla_{\mathbf{z}_{1:M}} \log(p(\mathbf{z}_{1:M}))$ as the score model and train $s_\theta(\mathbf{z}_{1:M})$ using score matching (Hyvärinen & Dayan, 2005). In practice, the score matching objective does not scale to the large dimension which is required in our setup, and other alternatives such as denoising score-matching (Vincent, 2011) and sliced score-matching (Song et al., 2020a) have been proposed.

Here we use denoising score matching in the continuous form that diffuses the latent samples into noise distribution and can be written in a stochastic differential equation (SDE) form (Song et al., 2020b). The forward process will be an SDE of $d\mathbf{z} = f(\mathbf{z}, t)dt + g(t)d\mathbf{w}$ where $\mathbf{w}$ is the Brownian motion. To reverse the noise distribution back to the latent distribution, we use the trained score model and sample iteratively from the reverse SDE $d\mathbf{z} = \left[\mathbf{f}(\mathbf{z}, t) - g(t)^2\nabla_{\mathbf{z}} \log p_t(\mathbf{z})\right] dt + g(t)d\overline{\mathbf{w}}$. The score network which approximates $\nabla_{\mathbf{z}} \log p_t(\mathbf{z})$ is trained using eq. 4. We use the unweighted version where $\lambda(t) = 1$.

$$\boldsymbol{\theta}^* = \arg\min_{\boldsymbol{\theta}} \mathbb{E}_t \left\{ \lambda(t)\mathbb{E}_{\mathbf{z}(0)}\mathbb{E}_{\mathbf{z}(t)|\mathbf{z}(0)} \left[\left\|\mathbf{s}_{\boldsymbol{\theta}}(\mathbf{z}(t), t) - \nabla_{\mathbf{z}(t)} \log p_{0t}(\mathbf{z}(t) \mid \mathbf{z}(0))\right\|_2^2\right]\right\} \tag{4}$$

## 2.1 Inference with missing modalities

The goal of inference is to sample unobserved modalities (indexed by $\mathbf{u} \subseteq \{1, \cdots, M\}$) given the observed modalities (indexed by $\mathbf{o} \subset \{1, \cdots, M\}$) from $p(\mathbf{x}_\mathbf{u}|\mathbf{x}_\mathbf{o})$. We define a variational lower bound on log-probability $\log p(\mathbf{x}_\mathbf{u}|\mathbf{x}_\mathbf{o})$ using posterior $q(\mathbf{z}_\mathbf{u}|\mathbf{x}_\mathbf{o})$ on the latent variables of the unobserved modalities $\mathbf{z}_\mathbf{u}$:

$$\begin{aligned}
\log p(\mathbf{x}_\mathbf{u}|\mathbf{x}_\mathbf{o}) &= \log \mathbb{E}_{q(\mathbf{z}_\mathbf{u}|\mathbf{x}_\mathbf{o})} p(\mathbf{x}_\mathbf{u}|\mathbf{x}_\mathbf{o}, \mathbf{z}_\mathbf{u}) \\
&\geq \mathbb{E}_{q(\mathbf{z}_\mathbf{u}|\mathbf{x}_\mathbf{o})} \log p(\mathbf{x}_\mathbf{u}|\mathbf{x}_\mathbf{o}, \mathbf{z}_\mathbf{u}) \\
&= \mathbb{E}_{q(\mathbf{z}_\mathbf{u}|\mathbf{x}_\mathbf{o})} \log p(\mathbf{x}_\mathbf{u}|\mathbf{z}_\mathbf{u}),
\end{aligned} \tag{5}$$

where $p(\mathbf{x}_\mathbf{u}|\mathbf{z}_\mathbf{u}) = \prod_{k\in\mathbf{u}} p_{\psi_k}(\mathbf{x}_k|\mathbf{z}_k)$ and $p_{\psi_k}(\mathbf{x}_k|\mathbf{z}_k)$ is the generative model for modality $k$. We define the posterior distribution $q(\mathbf{z}_\mathbf{u}|\mathbf{x}_\mathbf{o})$ as the following:

$$q(\mathbf{z}_\mathbf{u}|\mathbf{x}_\mathbf{o}) = \sum_{\mathbf{z}_\mathbf{o}} \left[\prod_{i\in\mathbf{o}} q_{\phi_i}(\mathbf{z}_i|\mathbf{x}_i)\right] p_\theta(\mathbf{z}_\mathbf{u}|\mathbf{z}_\mathbf{o}), \tag{6}$$

where $q_{\phi_k}(\mathbf{z}_k|\mathbf{x}_k)$ is the recognition model for modality $k$.

In order to sample from $q(\mathbf{z}_\mathbf{u}|\mathbf{x}_\mathbf{o})$, following eq. 6, we first sample $\mathbf{z}_\mathbf{o}$ for all observed modalities, and then sample $\mathbf{z}_\mathbf{u}$ from $p_\theta(\mathbf{z}_\mathbf{u}|\mathbf{z}_\mathbf{o})$ by fixing $\mathbf{z}_\mathbf{o}$ which is the latent representation of the observed modalities and updating the unobserved ones during sampling. When all modalities are missing, we will update all modalities in the sampling step.

After running the sampling for $T$ steps we use the generative models $p(\mathbf{x}_\mathbf{u}|\mathbf{z}_\mathbf{u}) = \prod_{k\in\mathbf{u}} p_{\psi_k}(\mathbf{x}_k|\mathbf{z}_k)$ to sample the unobserved modalities. We use the Predictor-Corrector (PC) sampling algorithm (Song et al., 2020b) which is a mix of Euler-Maruyama and Langevin Dynamics (Welling & Teh, 2011) to sample from the score model.

## 2.2 Coherence guidance

### 2.2.1 Energy-based coherence guidance

The score-based model improves the coherence among predicted modalities, however, when the number of unobserved modalities increases, it is more likely that the predicted modalities are not aligned with the

observed modalities. To address this issue, we use extra conditional guidance during the reverse process for sampling (Dhariwal & Nichol, 2021). In particular, we train an energy-based model that assigns low energy to a coherent pair of modalities and high energy to incoherent ones. We define the energy-based model $E_\omega(\mathbf{z}_o, \mathbf{z}_u)$ over the latent representations of an observed modality $\mathbf{z}_o$ and an unobserved modality $\mathbf{z}_u$. Here we assume that the latent representation of all modalities has the same dimension but in general, it can work with any dimension. During training, we randomly select two modalities $(\mathbf{x}_o, \mathbf{x}_u)$ as a positive pair and substitute the second modality with an incoherent example $\mathbf{x}_n$ in training data to construct a negative pair $(\mathbf{x}_o, \mathbf{x}_n)$. We train $E_\omega$ using noise contrastive estimation (Gutmann & Hyvärinen, 2010):

$$\max_\omega \; \mathbb{E}_{(x_o,x_u)\sim p(x_o,x_u),z_o\sim q(z|x_o),z_u\sim q(z|x_u)} \log \frac{1}{1+\exp\left(E_\omega\left((z_o,z_u)\right)\right)} \tag{7}$$

$$+ \mathbb{E}_{x_o\sim p(x_o),x_n\sim p(x_n),z_o\sim q(z|x_o),z_n\sim q(z|x_n)} \log \frac{1}{1+\exp\left(-E_\omega(z_o,z_n)\right)}$$

We also perturb the latent representations with the same perturbation kernel used to train the score model. The final score function with coherence guidance has the following form:

$$\tilde{s}(z_u) = s_\theta(z_u) - \gamma \nabla_{z_u} E_\omega(z_o, z_u), \tag{8}$$

where $\gamma$ controls the strength of guidance. During each step of inference, we randomly select $x_o$ from observed modalities and randomly select $x_u$ from unobserved modalities and update the score of $z_u$ using eq. 8.

### 2.2.2 Contrastive guidance

Another way to enhance the conditional performance of the score-based model is to leverage contrastive guidance from an auxiliary encoder network $E(\mathbf{x})$. This encoder maps the input data $\mathbf{x}$ from all modalities to a common embedding space, enabling conditional alignment through contrastive learning objectives (Rombach et al., 2021; Tang et al., 2023). The resulting embeddings capture semantic information that can guide the score-based model during the sampling process.

Specifically, we incorporate secondary encoder embeddings $E(\mathbf{x})$ as an additional conditioning input to the score model $\mathbf{s}_\theta(\mathbf{z}(t), t, E(\mathbf{x}))$. The model can use this aligned representation which will be useful in providing guidance. We show this method is supportive as an additional guidance method on the datasets in the experiments section. Compared to the energy-based coherence guidance, which is trained independently of the score model and added during sampling time, this method requires the secondary encoders to be present when training the score model. We train the secondary encoders, $E(\mathbf{x})$, using contrastive learning similar to Radford et al. (2021) where the similarity between representations from the same pairs of data are maximized and the similarity between representations from unrelated pairs are minimized.

## 3 Related Works

Our work is heavily inspired by earlier works on deep multimodal learning (Ngiam et al., 2011; Srivastava & Salakhutdinov, 2014). Ngiam et al. (2011) use a set of autoencoders for each modality and a shared representation across different modalities and trained the parameters to reconstruct the missing modalities given the present one. Srivastava & Salakhutdinov (2014) define deep Boltzmann machine to represent multimodal data with modality-specific hidden layers followed by shared hidden layers across multiple modalities, and use Gibbs sampling to recover missing modalities.

Suzuki et al. (2017) approached this problem by maximizing the joint ELBO and additional KL terms between the posterior of the joint and the individuals to handle missing data. Tsai et al. (2019) propose a factorized model in a supervised setting over model-specific representation and label representation, which capture the shared information. The proposed factorization is $q(\mathbf{z}_{1:M}, \mathbf{z}_y|\mathbf{x}_{1:M}, \mathbf{y}) = q(\mathbf{z}_y|\mathbf{x}_{1:M}) \prod_{k=1}^{M} q(\mathbf{z}_k|\mathbf{x}_k)$, where $\mathbf{z}_y$ is the latent variable corresponding to the label.

Most current approaches define a multimodal variational lower bound similar to variational autoencoders Kingma & Welling (2014) using a shared latent representation for all modalities:

$$\log p(\mathbf{x}_{1:M}) \geq \mathbb{E}_{q_\phi(\mathbf{z}|\mathbf{x}_{1:M})}\left[\log p_\psi(\mathbf{x}_{1:M}|z)\right] - D_{\mathrm{KL}}(q_\phi(z|\mathbf{x}_{1:M})\|p(\mathbf{z})). \tag{9}$$

Similar to our setup $p_\psi(\mathbf{x}_{1:M}|z) = \prod_k p_{\psi_k}(\mathbf{x}_k|\mathbf{z})$, but $q_\phi(\mathbf{z}|\mathbf{x}_{1:M})$ is handled differently. Wu & Goodman (2018) use a product of experts to describe $q$: $q_{\mathrm{PoE}}(\mathbf{z}|\mathbf{x}_{1:M}) = p(\mathbf{z})\prod_{k=1}^M q_{\phi_k}(\mathbf{z}|\mathbf{x}_k)$. Assuming $p(\mathbf{z})$ and each of $q_{\phi_k}(\mathbf{z}|\mathbf{x}_k)$ follow a Gaussian distribution, $q_{\mathrm{PoE}}$ can be calculated in a closed form, and we can optimize the multimodal ELBO accordingly. To get a good performance on generating missing modality, Wu & Goodman (2018) sample different ELBO combinations of the subset of modalities. Moreover, the sub-sampling proposed by Wu & Goodman (2018) results in an invalid multimodal ELBO (Wu & Goodman, 2019). The MVAE proposed by (Wu & Goodman, 2018) generates good-quality images but suffers from low cross-modal coherence. To address this issue Shi et al. (2019) propose constructing $q_\phi(\mathbf{z}|\mathbf{x}_{1:M})$ as a mixture of experts: $q_{\mathrm{MoE}}(\mathbf{z}|\mathbf{x}_{1:M}) = \frac{1}{M}\sum_k q_{\phi_k}(\mathbf{z}|\mathbf{x}_k)$. However, as pointed out by Daunhawer et al. (2022), sub-sampling from the mixture component results in lower generation quality, while improving the coherence.

Sutter et al. (2021) propose a mixture of the product of experts for $q$ by combining these two approaches: $q_{\mathrm{MoPoE}}(\mathbf{z}|\mathbf{x}_{1:M}) = \frac{1}{2^M}\sum_{\mathbf{x_m}} q(\mathbf{z}|\mathbf{x_m})$, where $q(\mathbf{z}|\mathbf{x_m}) = \prod_{k\in\mathbf{m}} q(\mathbf{z}|\mathbf{x}_k)$ and $\mathbf{m}$ is a subset of modalities. The number of mixture components grows exponentially as the number of modalities increases. MoPoE has better coherence than PoE, but as discussed by Daunhawer et al. (2022) sub-sampling modalities in mixture-based multimodal VAEs result in loss of generation quality of the individual modalities. To address this issue, more recently and in parallel to our work, Palumbo et al. (2023) introduce modality-specific latent variables in addition to the shared latent variable. In this setting the joint probability model over all variables factorizes as $p(\mathbf{x}_{1:M},\mathbf{z},\mathbf{w}_{1:M}) = p(\mathbf{z})\prod_k p(\mathbf{x}_k|\mathbf{z},\mathbf{w}_k)p(\mathbf{w}_k)$ and $q$ factorizes as $q_{\mathrm{MMVAE+}}(\mathbf{z},\mathbf{w}_{1:M}|\mathbf{x}_{1:M}) = q_{\phi_z}(\mathbf{z}|\mathbf{x})\prod_k q_{\phi_k}(\mathbf{w}_k|\mathbf{x}_k)$. Using modality-specific representation is also explored by Lee & Pavlovic (2021), however, the approach proposed by Palumbo et al. (2023) is more robust to controlling modality-specific representation vs shared representation. But since the shared component is a mixture of experts of individual components, it can only use one of them at a time during inference which limits its ability to use additional observations that are available as the number of given modalities increase.

Sutter et al. (2020) propose an updated multimodal objective that consists of a JS-divergence term instead of the normal KL term with a mixture-of-experts posterior. They also add additional modality-specific terms and a dynamic prior to approximate the unimodal and the posterior term. Though these additions provide some improvement, there is still a need for a model that balances coherence with quality (Palumbo et al., 2023).

Wolff et al. (2022) propose a hierarchical multimodal VAEs for the task where a generative model of the form $p(x,g,z)$ and an inference model $q(g,z|x_{1:M})$ containing multiple hierarchies of $z$ where $g$ holds the shared structures of multiple modalities in a mixture-of-experts form $q(g|x_{1:M})$. They argue that the modality-exclusive hierarchical structure helps in avoiding the modality sub-sampling issue and can capture the variations of each modality. Though the hierarchy gives some improvement in results, the model is still restricted in capturing the shared structure in $g$ discussed in their work.

Suzuki & Matsuo (2023) introduce a multimodal objective that avoids sub-sampling during training by not using a mixture-of-experts posterior to avoid the main issue discussed by Daunhawer et al. (2022). They propose a product-of-experts posterior multimodal objective with additional unimodal reconstruction terms to facilitate cross-modal generations.

Hwang et al. (2021) propose an ELBO that is derived from an information theory perspective that encourages finding a representation that decreases the total correlation. They propose another ELBO objective which is a convex combination of two ELBO terms that are based on conditional VIB and VIB. The VIB term decomposes to ELBOs of previous approaches. The conditional term decreases the KL between the joint posterior and individual posteriors. They use a product-of-experts as their joint posterior.

Finally, parallel to our work, Xu et al. (2023) propose diffusion models to tackle the multimodal generation which uses multi-flow diffusion with data and context layer for each modality and a shared global layer. Also, concurrent to our work, Bounoua et al. (2024) proposed a latent diffusion model for multimodal data using a

score-based diffusion model. They use autoencoders to project the modalities into latent space and train a joint model similar to ours while training the score model differently by masking some of the modalities in the diffusion process to facilitate conditional performance. Tang et al. (2023) introduced any-to-any generation, but in contrast to our work, their approach involves training a separate latent diffusion model for each modality. Diffusion models are then conditioned on an aligned latent space (they have used text modality as the anchor) for conditional generation and are trained jointly to offer joint modality prediction. Our approach is based on learning a joint latent space and avoiding alignment or learning common latent space, while their approach is based on learning an aligned latent space for conditioning independent modality generation, which is highly dependent on finding the aligned latent space for different modalities.

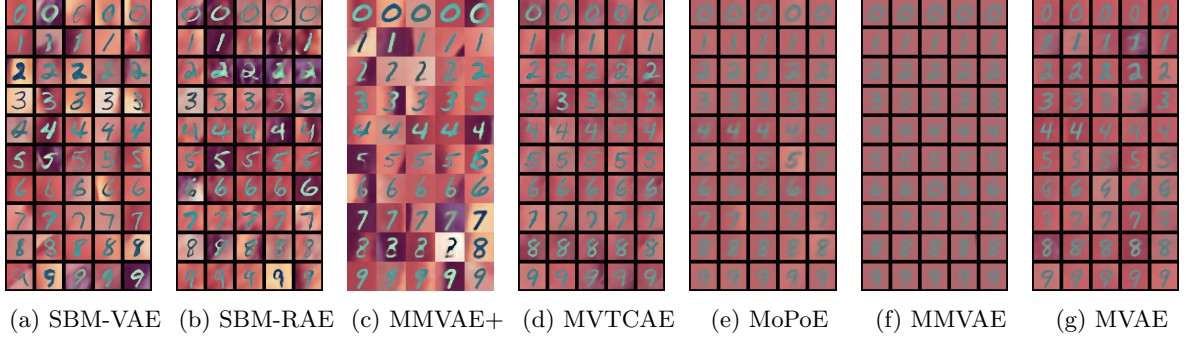

(a) SBM-VAE  (b) SBM-RAE  (c) MMVAE+  (d) MVTCAE  (e) MoPoE  (f) MMVAE  (g) MVAE

Figure 3: Multiple conditionally generated samples for each digit from the third modality. Each column shows samples, from 0 to 9, generated conditionally given the remaining modalities.

## 4    Experiments

We study our proposed methods and selected baselines using an extended version of PolyMNIST (Sutter et al., 2021) as well as high-dimensional CelebAMask-HQ (Lee et al., 2020) datasets. Extended PolyMNIST is ideal for observing the performance patterns in the presence of a different number of modalities, which is also used by previous works, while CelebAMask-HQ tests the behavior of the methods in a high-dimensional setting. The experiments on the CelebAMask-HQ demonstrate that multimodal VAEs, in general, can be used to solve structured prediction problems. Finally, we study the robustness of multimodal VAEs in the presence of adversarial attacks. Other experiments including an audio modality are available in the Appendix section A.6.

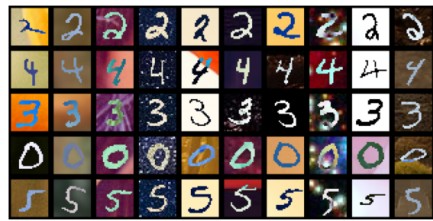

Figure 2: Extended PolyMnist Dataset

### 4.1    Extented PolyMnist

The original PolyMNIST introduced by Sutter et al. (2021) has five modalities and in order to study the behavior of the methods on a larger number of modalities, we extended the number of modalities to ten. Figure 2 shows samples of the Extended PolyMNIST data.

We compare our methods SBM-VAE and SBM-RAE, which substitute individual variational autoencoder with a regularized deterministic autoencoder (Ghosh et al., 2020); see Appendix A.1 for the details of SBM-RAE, as well as their counterparts with energy-based coherence guidance SBM-VAE-C and SBM-RAE-C plus SBM-VAE-T which uses contrastive guidance with baselines of MVAE (Wu & Goodman, 2018), MMVAE (Shi et al., 2019), MoPoE (Sutter et al., 2021), MVTCAE (Hwang et al., 2021), and MMVAE+ (Palumbo et al., 2023). We added SBM-RAE to show that our approach can be used with any auto-encoder structure and is not limited to only a variational auto encoder, unlike the baselines.

Hyperparameters and neural network design are discussed in detail in Appendix A.2. Conditional generation is done for MVAE and MVTCAE using PoE of the posteriors of the given modalities, and for the mixture models, a mixture component is chosen uniformly from the observed modalities.

We evaluate all methods on both prediction coherence and generative quality. To measure the coherence, we use a pre-trained classifier to extract the label of the generated output and compare it with the associated label of the observed modalities Shi et al. (2019). The coherence of the unconditional generation is evaluated by counting the number of consistent predicted labels from the pre-defined classifier. We also measure the generative quality of the generated modalities using the FID score (Heusel et al., 2017). All the results that are shown are run for at least 3 times and the mean is shown. The standard deviation is shown as shades under the curves in each figure. Figure 3 shows the generated samples from the third modality given the rest. The SBM models generate high-quality images with considerable variation, very close to the original data, while most of the baselines generate more blurry images with a lower variation.

**Unconditional generation**. We first study unconditional generation. Figure 4 shows the generated samples using SBM-VAE. For the full sample sets for all baseline methods please see Figure 26 in Appendix. We evaluate unconditional coherence by counting the number of consistent labels across different modalities after classifying the generated images using a pre-trained classifier. In other words, the unconditionally generated images should represent the same number expressed in the different modalities. Since no modalities have been observed, we don't use coherence guidance for unconditional generation. The results have been reported in Figure 5. Approximately 71% of the generated images using SBM models are coherent which is considerably better than the other baselines. The coherence percentage gap between SBM models and the next best model, MMVAE, is much larger for nine out of ten, 92.5% vs 74.9%, respectively. To ensure that our models' high performance is not due to mode drop, we generate 10000 unconditional samples and calculate the mode coverage for all digits. Our methods and MMVAE+ uniformly cover all the models, while MVTCAE coverage is not uniform. See Figure 20 in Appendix for mode coverage.

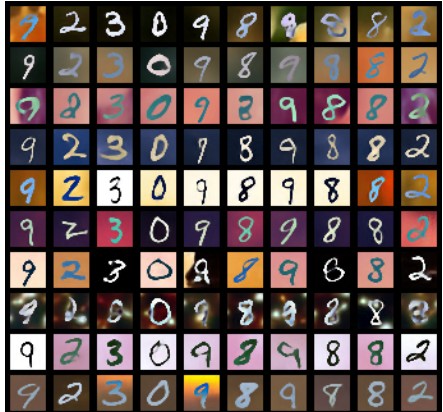

Figure 4: Unconditional samples from 10 modalities using SBM-VAE. Columns represent different samples from each modality.

Further, we study the scalability of methods by changing the number of trained modalities from two to ten and then evaluating unconditional FID. Similar to unconditional coherence, for unconditional FID we don't use coherence guidance. The outcome is shown in Figure 6. The performance of our proposed methods is consistent and superior in the presence of different numbers of data modalities.

**Conditional generation**. We run similar experiments to study the behavior of the methods for conditional generation. In the conditional generation, we assume we are generating the first modality given the rest. The selection of the first modality is to remain consistent as we increase the number of modalities from two to ten. Figure 7 shows the outcome. The SBM models achieve the best performance in generation quality in both conditional and unconditional generations as we scale the number of modalities. This shows that our approach can scale to a high number of modalities without compromise. Specifically, the SBM-VAE variants show consistent FID and coherence, for which the complexity of the sampling from prior remains the same as we increase the number of data modalities.

We also study, how the conditional coherence and FID change with the number of the observed modality in a single model trained with 10 modalities. For an accurate conditional model, we expect as we observe more modalities, the accuracy of conditional generation will improve without a decrease in quality. Figure 8 shows the FID score of the last modality as we increase the given modalities and Figure 9 demonstrates the conditional coherence of the predicted images on the last modality given a different number of observed modalities. Our models show consistent FID while also having increasing accuracy. The figure shows the

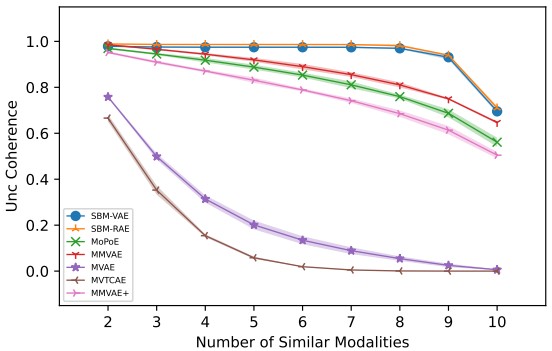

Figure 5: Unconditional coherence. The x-axis shows the number of coherent modalities and the y-axis shows the percentage of such coherent predicted modalities in the generated output

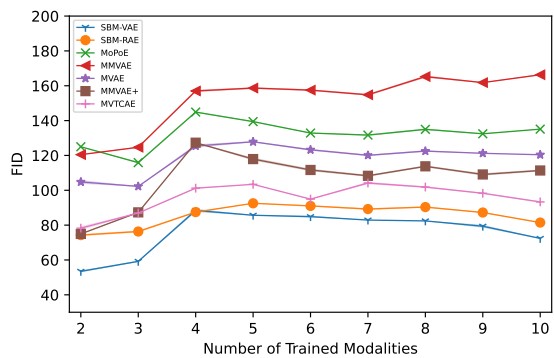

Figure 6: Average unconditional FID as the number of modalities the models are trained on increase as shown on the x-axis

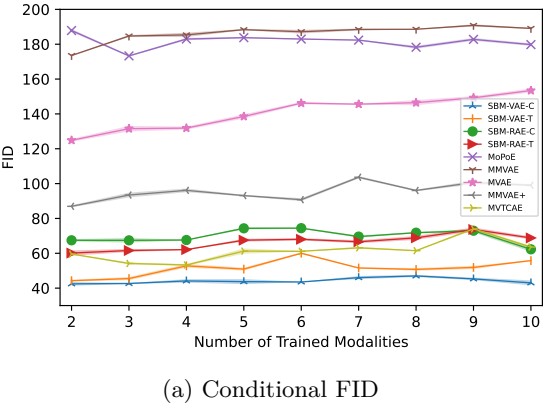

(a) Conditional FID

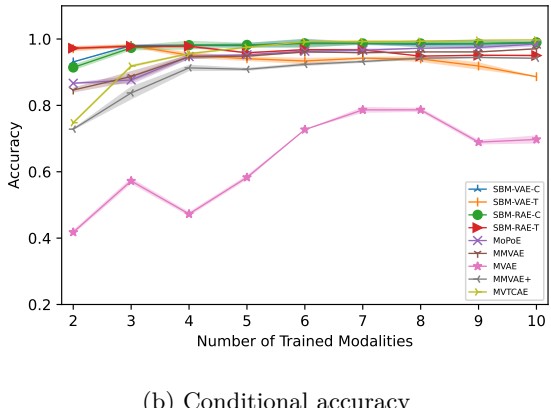

(b) Conditional accuracy

Figure 7: a) Conditional FID and b) Conditional accuracy of the generated first modality given the rest of modalities as the number of modalities the models are trained on increases as shown on the x-axis.

deterioration in quality quantified by increasing FID as more modalities are observed for the compared methods except for the SBM models and MMVAE+. SBM-based multimodal VAEs and MMVAE+ both are equipped with a way to capture modality-specific representation, so increasing the number of observed modalities does not affect their generation quality. The SBM exploits the additional modalities to generate better and more accurate images as we increase the number of modalities without losing quality. Similar experiments have been reported for predicting the first modality in the Appendix section (see Figure 12 and Figure 13).

The plain score model suffers from low coherence when we have only one or two observed modalities out of the ten. But with the help of the coherence guidance, we can guide the model towards the correct output and that helps increase the coherence of SBM-VAE and SBM-RAE when the number of observed modalities is low. In addition to that, increasing the number of sampling steps from 100 to 1000 also helps when a few modalities are observed. We demonstrate the effect of coherence guidance and the number of inference steps in Figure 11. The SBM-VAE-T model also shows a high coherence even when the number of observed modalities is low. Further details as well as more generated samples and more results on other modalities have been reported in Appendix A.4.

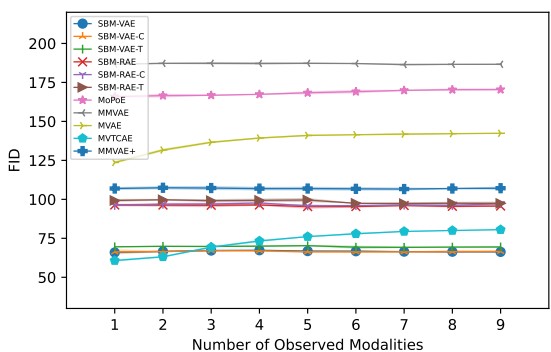

Figure 8: The conditional FID of the last modality generated by incrementing the given modality at a time. The x-axis shows how many modalities are given to generate the modality and the y-axis shows the FID score of the generated modality.

Figure 9: The conditional accuracy of the last modality generated by incrementing the given modality at a time. The x-axis shows how many modalities are given to generate the modality and the y-axis shows the accuracy of the generated modality.

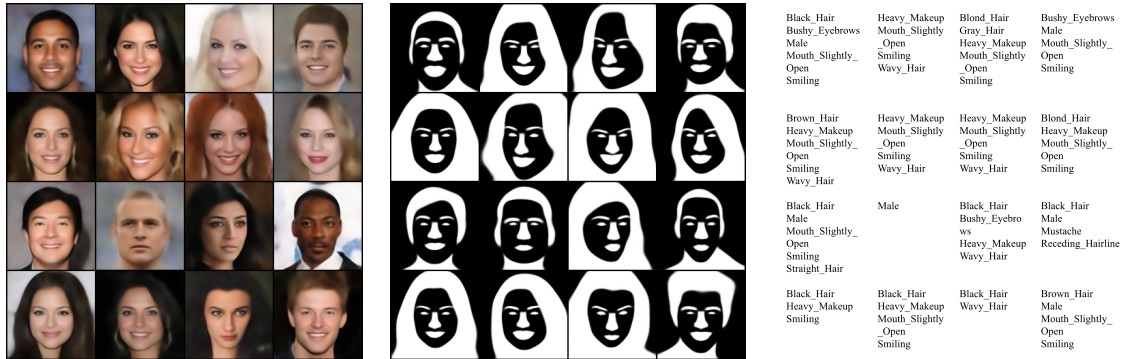

Figure 10: Unconditional generation using SBM-VAE, Left) images, Center) masks, and Right) attributes.

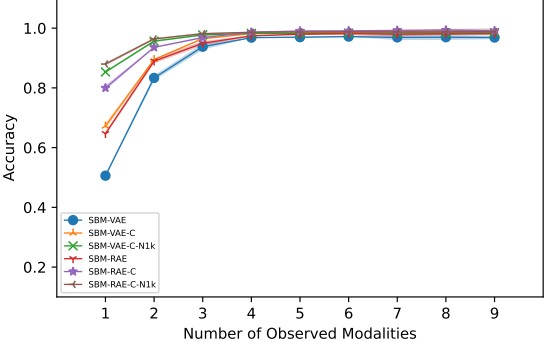

Figure 11: Effect of having the energy-based coherence guidance model and number of sampling steps

## 4.2 CelebAMask-HQ

The images, masks, and attributes of the CelebAMask-HQ dataset can be treated as different modalities of expressing the visual characteristics of a person. A sample from the dataset is shown in Figure 1. The images and masks are resized to 128 by 128. The masks are either white or black where all the masks given

in the CelebAMask-HQ except the skin mask are drawn on top of each other as one image. We follow the pre-processing of Wu & Goodman (2018) by selecting 18 out of 40 existing attributes as described.

We compare SBM-VAE, SBM-VAE-C, SBM-VAE-T, SBM-RAE, SBM-RAE-C, SBM-RAE-T from the SBM variants and MoPoE by Sutter et al. (2021), MVTCAE by Hwang et al. (2021), MMVAE+ by Palumbo et al. (2023), and MLD by Bounoua et al. (2024) as baselines. See Appendix A.9 for the experimental setups of these methods.

We evaluate the generation quality of the image modality using FID score and generation accuracy of mask and attribute modalities using sample-average $F_1$ score. Table 1 shows how our methods compare with the baselines in the presence of one or two observed modalities. We have also reported the performance of a supervised model trained to predict attributes and masks directly given the images. Table 2 shows the unconditional image generation performance of the models. SBM-VAE and SBM-RAE variants generate high-quality images compared to the baselines in both conditional and unconditional settings, while on the mask modalities, MoPoE and MVTCAE achieve a better $F_1$ score on mask prediction. We can note that the contrastive guidance is very helpful and the top-performing model when predicting the attributes conditionally. We can also observe that conditional guidance consistently improves the coherence of score-based models. The SBM models achieve superior performance when predicting the image modality consistently and also perform comparatively or better in the other modalities, showing the effectiveness of the method in different modalities.

Finally, we show the unconditional generation across different modalities using SBM-VAE in Figure 10. Please see Appendix A.10 for more experiments and qualitative samples on the CelebAMask-HQ dataset.

### 4.2.1 Robustness to Adversarial Attacks

In this section, we study how multimodal VAEs and SBM-VAEs perform in the presence of a blackbox adversarial attack. We choose blackbox attack because the inference algorithm for SBM is not end-to-end differentiable. We apply the blackbox method by Papernot et al. (2017) to generate the adversarial examples using a supervised model that classifies the attributes from the images of the CelebAHQ dataset and then feed these perturbed images to both multimodal VAEs based on MoPoE, MVTCAE, MMVAE+ as well as our SBM-VAE-C and the contrastive guided SBM-RAE-T. The perturbed images are created using the Fast Gradient Sign Method (FGSM) Goodfellow et al. (2015) with $\epsilon=0.05$. We then use the adversarial images instead of the original images and predict the attributes.

As reported in Table 3, SBMVAE-C is much less affected by the adversarial attacks as its performance decreased by approximately 4% while MoPoE's and MVTCAE's performance dropped by 19.1% and 13.7%, respectively. This shows that our method is more robust to blackbox adversarial attacks. The robustness of SBM-based VAEs is because of the inference procedure which causes purification Nie et al. (2022) on the latent representation of the perturbed modality, by mapping back the noisy latent representation to the original latent manifold. On the other hand, SBM-RAE-T is affected the most as the secondary conditioning encoder directly takes the images which will be affected when given an adversarial input.

## 5 Conclusion and Discussion

Multimodal VAEs are an important tool for modeling multimodal data. In this paper, we provide a different multimodal posterior using score-based models. Our proposed method learns the correlation of latent spaces of unimodal representations using a joint score model in contrast to the traditional multimodal VAE ELBO. We show that our method can generate better-quality samples and, at the same time, preserve coherence among modalities. We have also shown that our approach is scalable to multiple modalities. We also introduce coherence guidance to improve the coherence of the generated modalities. The coherence guidance effectively improves the coherence of our score-based models. In conclusion, while our score-based multimodal VAE approach offers certain advantages, it has a trade-off. Unlike traditional multimodal VAEs, our model requires a computationally expensive sampling procedure during inference. This ultimately forces a balance between the quality and coherence of predictions and the computational resources needed. This limitation highlights an area for future research. **Broader Impact:** We believe this model can be used in real-world applications

Table 1: CelebAMask-HQ generation performance for each modality. Image quality is measured using FID score, and the quality of Mask and Attribute modalities are measured using $F_1$ score. The second row indicates the observed modalities.

| | Attribute | | Image | | | Mask | |
|---|---|---|---|---|---|---|---|
| GIVEN | Both | Img | Both | Mask | Attr | Both | Img |
| | F1 | F1 | FID | FID | FID | F1 | F1 |
| SBM-RAE | $0.62_{(\pm 0.003)}$ | $0.6_{(\pm 0.004)}$ | $84.9_{(\pm 0.19)}$ | $86.4_{(\pm 0.02)}$ | $85.6_{(\pm 0.48)}$ | $0.83_{(\pm 0.001)}$ | $0.82_{(\pm 0.001)}$ |
| SBM-RAE-C | $0.66_{(\pm 0.003)}$ | $0.64_{(\pm 0.004)}$ | $83.6_{(\pm 0.05)}$ | $82.8_{(\pm 0.05)}$ | $83.1_{(\pm 0.14)}$ | $0.83_{(\pm 0.004)}$ | $0.82_{(\pm 0.004)}$ |
| SBM-RAE-T | $\mathbf{0.77}_{(\pm 0.001)}$ | $\mathbf{0.77}_{(\pm 0.001)}$ | $83.1_{(\pm 0.09)}$ | $82.6_{(\pm 0.45)}$ | $80.2_{(\pm 0.34)}$ | $0.84_{(\pm 0.001)}$ | $0.84_{(\pm 0.001)}$ |
| SBM-VAE | $0.62_{(\pm 0.003)}$ | $0.58_{(\pm 0.005)}$ | $81.6_{(\pm 0.10)}$ | $81.9_{(\pm 0.03)}$ | $78.7_{(\pm 0.25)}$ | $0.83_{(\pm 0.001)}$ | $0.83_{(\pm 0.001)}$ |
| SBM-VAE-C | $0.69_{(\pm 0.005)}$ | $0.66_{(\pm 0.001)}$ | $82.4_{(\pm 0.1)}$ | $\mathbf{81.7}_{(\pm 0.29)}$ | $76.3_{(\pm 0.7)}$ | $0.84_{(\pm 0.02)}$ | $0.84_{(\pm 0.001)}$ |
| SBM-VAE-T | $0.74_{(\pm 0.001)}$ | $0.74_{(\pm 0.003)}$ | $\mathbf{80.9}_{(\pm 0.14)}$ | $83.0_{(\pm 0.22)}$ | $\mathbf{73.8}_{(\pm 0.12)}$ | $0.84_{(\pm 0.001)}$ | $0.84_{(\pm 0.001)}$ |
| MLD | $0.71_{(\pm 0.005)}$ | $0.67_{(\pm 0.006)}$ | $81.7_{(\pm 0.25)}$ | $82.4_{(\pm 0.15)}$ | $80.29_{(\pm 0.6)}$ | $0.86_{(\pm 0.001)}$ | $0.86_{(\pm 0.001)}$ |
| MoPoE | $0.68_{(\pm 0.002)}$ | $0.71_{(\pm 0.004)}$ | $114.9_{(\pm 0.32)}$ | $101.1_{(\pm 0.16)}$ | $186.7_{(\pm 0.28)}$ | $0.85_{(\pm 0.002)}$ | $\mathbf{0.92}_{(\pm 0.001)}$ |
| MVTCAE | $0.71_{(\pm 0.001)}$ | $0.69_{(\pm 0.004)}$ | $94_{(\pm 0.45)}$ | $84.2_{(\pm 0.32)}$ | $87.2_{(\pm 0.08)}$ | $\mathbf{0.89}_{(\pm 0.001)}$ | $0.89_{(\pm 0.003)}$ |
| MMVAE+ | $0.64_{(\pm 0.003)}$ | $0.61_{(\pm 0.002)}$ | $133_{(\pm 14.28)}$ | $97.3_{(\pm 0.04)}$ | $153_{(\pm 0.49)}$ | $0.82_{(\pm 0.003)}$ | $0.89_{(\pm 0.002)}$ |
| Supervised | | $0.79_{(\pm 0.001)}$ | | | | | $0.94_{(\pm 0.001)}$ |

Table 2: Unconditional performance on the CelebAMask-HQ dataset

| | SBM-VAE | SBM-RAE | MLD | MoPoE | MVTCAE | MMVAE+ |
|---|---|---|---|---|---|---|
| FID | $\mathbf{79.1}_{(\pm 0.07)}$ | $84.2_{(\pm 0.25)}$ | $82.8_{(\pm 0.08)}$ | $164.8_{(\pm 0.62)}$ | $162.2_{(\pm 1.08)}$ | $103.7_{(\pm 0.61)}$ |

Table 3: Performance on Adversarial Attacks by predicting the attributes given image from CelebAMask-HQ dataset

| Model | F1 with Adv Inputs | Percentage drop |
|---|---|---|
| MoPoE | $0.574_{(\pm 0.001)}$ | 19.1 |
| MVTCAE | $0.595_{(\pm 0.002)}$ | 13.7 |
| MMVAE+ | $0.576_{(\pm 0.001)}$ | 5.6 |
| SBM-RAE-T | $0.490_{(\pm 0.001)}$ | 36.3 |
| SBM-VAE-C | $\mathbf{0.634}_{(\pm 0.002)}$ | $\mathbf{3.9}$ |

because of its advantages and improve diverse applications like healthcare and other areas that rely on multiple modalities.

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

## A  Appendix

### A.1  SBM-RAE

Regularized autoencoders (RAEs) (Ghosh et al., 2020) can be used instead of VAEs in our setting. RAEs assume a deterministic encoder and regularize the latent space directly by penalizing $L_2$ norm of the latent variables:

$$\mathcal{L}_{\text{RAE}} = \mathcal{L}_{\text{REC}} + \beta||\mathbf{z}||_2^2 + \lambda\mathcal{L}_{\text{REG}}, \tag{10}$$

where $L_{\text{REC}}$ is the reconstruction loss for deterministic autoencoder and $L_{\text{REG}}$ is the decoder regularizer. In our setup, we don't use any decoder regularizer.

In order to generate a sample from the latent space, RAEs require to fit separate density estimators on the latent variables. In our case, the score models are responsible for generating samples from the latent space, which makes RAE a compelling choice for our setup. RAEs are capable of learning more complex latent structures, and expressive generative models such as score models can effectively learn that structure to generate high-quality samples.

## A.2 Model Architectures and Experimental setups for Extended PolyMNIST

The extended PolyMnist dataset was updated from the original PolyMnist dataset by Sutter et al. (2020) with different background images and ten modalities. It has 50,000 training set, 10,000 validation set, and 10,000 test set. The VAEs for each modality are trained with an initial learning rate of 0.001 using a $\beta$ value of 0.1 where all the prior, posterior, and likelihood are Gaussians. This also applies to all multimodal VAEs except MMVAE+ which uses Laplace distribution instead of Gaussian. The RAE for each modality was trained using the mean squared error loss with the norm of $||\mathbf{z}||_2^2$ regularized by a factor of $10^{-5}$ and a Gaussian noise added to $z$ before feeding to the decoder with mean 0 and variance of 0.01 where the hyperparameter values were tuned using the validation set. The encoders and decoders for all models use residual connections to improve performance and are similar in structure to the architecture used in Daniel & Tamar (2021) except for MMVAE+ were we used the original model because the model's performance doesn't generalize to different neural net architecture. For MMVAE+, modality-specific and shared latent sizes are each 32 and the model was trained similarly to the code provided by the paper [2] with the IWAE estimator with K=1. The detailed architecture can be found by referring to the code that is attached. We used latent size of 64 for our models and MMVAE+ and we increased the latent dimension of the other baselines to $64 * n$ where $n$ is the number of modalities because they don't have any modality specific representation. We chose the best $\beta$ value for each model using the validation set. For MoPoE and MMVAE, $\beta$ of 2.5 was chosen, for MMVAE+ 5, for MVAE 1 and for MVTCAE 0.1.

The neural net of SBM-VAE and SBM-RAE is a UNET network where we resize the latent size to 8x8. For SBM-VAE, samples are taken from the posterior at training time and the mean of posterior is taken at inference time. For SBM-RAE, the **z**s are taken directly. We use a learning rate of 0.0002 with the Adam optimizer (Kingma & Ba, 2015). The detailed hyperparameters are shown in table 4. We use the VPSDE with $\beta_0 = 0.1$ and $\beta_1$=5 with $N = 100$ and the PC sampling technique with Euler-Maruyama sampling and langevin dynamics. For modalities less than 10, we use $\beta_0$ of 1, the others hyperparameters remain the same. The energy-based model is a simple MLP with the softplus activation. We follow equation 7 to train the models.

The classifier used for evaluation is composed of three convolutional layers followed by ReLU activations and two fully connected layers where the last one outputs 10 logits for classification. It's trained using the cross-entropy loss using the training data composed of all modalities and achieves an average accuracy 98.7 on the test set of all modalities.

Table 4: Score Hyperparameters PolyMnist

| Model | $\beta_{min}$ | $\beta max$ | N | LD per step | EBM-scale | batch-size |
|---|---|---|---|---|---|---|
| SBM 10mod | 0.1 | 5 | 100 | 1 | 1000 | 256 |
| SBM below 10mod | 1 | 5 | 100 | 1 | 1000 | 256 |

## A.3 Training and Inference Algorithm

We follow the Song et al. (2020b) to train the score models using the latent representation from the encoders. The following algorithms 1, 2 show the training and inference algorithm we use.

---

[2]MMVAE+ code is taken and updated from the official repo provided at https://github.com/epalu/mmvaeplus

---

**Algorithm 1** Training

---

**Require:** $M$, $N$,                                                      ▷ M - modality, N -epochs
    $\mathbf{z} = []$
    **for** $i = 1$ to $M$ **do**
        Get $\mathbf{x}_i$                                      ▷ Get the input x from modality i
        Sample $\mathbf{z}_i$ from $q(\mathbf{z}|\mathbf{x}_i)$                         ▷ Sample z from the encoder
        Append $\mathbf{z}_i$ to $\mathbf{z}$
    **end for**
    **for** $epoch = 1$ to $N$ **do**
        Diffuse $\mathbf{z}$
        Compute $\mathbf{s}_\theta(\mathbf{z}, t)$
        Calculate loss using equation 4
        Backpropagate and update the weights of the SBM
    **end for**

---

**Algorithm 2** Inference

---

    **if** Unconditional **then**
        Sample $\mathbf{z}$ from $\mathcal{N}(\mathbf{0}, \mathbf{I})$ and stack to get $\mathbf{Z}$
    **else if** Conditional **then**
        Sample $\mathbf{z}$ from $\mathcal{N}(\mathbf{0}, \mathbf{I})$ if missing
        Sample $\mathbf{z}$ from encoder $q(\mathbf{z}|\mathbf{x}_i)$ if present
        Stack all $\mathbf{z}$
    **end if**
    **for** $i = 1$ to $n$ **do**                                 ▷ n is number of sampling iterations
        Update missing $\mathbf{z}$s by the following equation using the PC sampling
        **if** use guidance **then**
            Add gradient from EBM to the score
        **end if**
    **end for**
    Feed the $\mathbf{z}$ to the respective decoder to get output

---

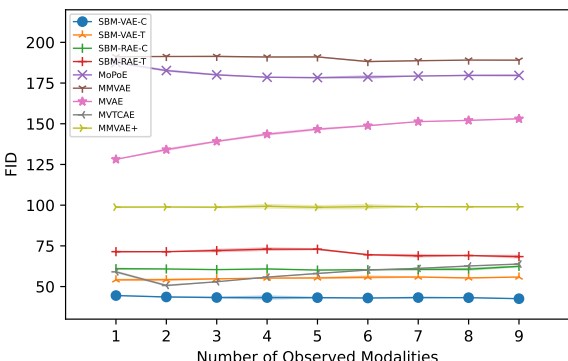
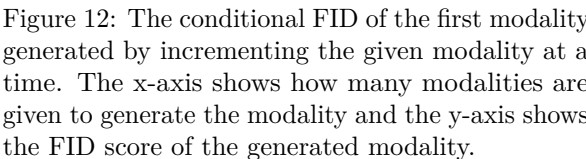

Figure 12: The conditional FID of the first modality generated by incrementing the given modality at a time. The x-axis shows how many modalities are given to generate the modality and the y-axis shows the FID score of the generated modality.

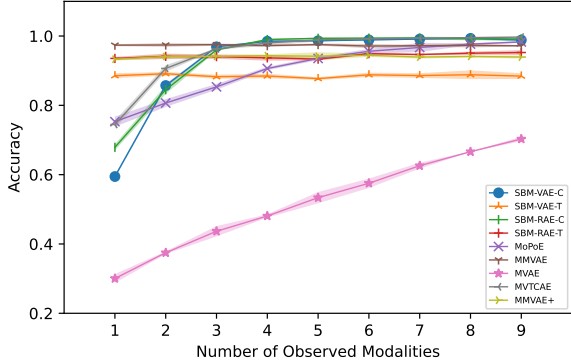

Figure 13: The conditional accuracy of the first modality generated by incrementing the given modality at a time. The x-axis shows how many modalities are given to generate the modality and the y-axis shows the accuracy of the generated modality.

### A.4 Ablation study

### A.4.1 Extended PolyMnist results

Here we show additional graphs for different modalities in addition to the ones shown in the main paper. Figure 12 shows the same setup as figure 8 in the main paper but for another modality.

We also show the average conditional coherence and average conditional FID for each model averaging over the modalities generated given the rest. Table 5 shows the result of that. Figure 14 and 15 show the results per modality.

### A.4.2 $\beta$ Ablation

In Table 6, we discuss how $\beta$ of the unimodal VAE affects the conditional FID and the conditional coherence of SBM-VAE. As the score model is trained on the samples from the posterior, samples that have good reconstruction are important for the score model to learn well. This is also evident in the result as lower $\beta$ values have better result than higher $\beta$ values as VAEs with lower $\beta$ values have better reconstruction. The score model also learns the gap created in unconditional sampling due to this trade-off as the score model transforms normal Gaussian noise to posterior samples during inference. The table shows a score model on the first two modalities of Extended PolyMnist trained using different $\beta$ values of 0.1,0.5, and 1 and their average conditional result.

In figure 16, 17, 19, and 16, we also show results of different $\beta$ values for MMVAE+ and MVTCAE predicting the last modality as the number of observed modalities is increased from 1 to 9.

### A.5 Mode coverage

In this section, in addition to FID, we discuss an evaluation technique for unconditional generation. We generate 10,000 unconditionally generated images from the multimodal Extended PolyMnist dataset and count how many digits are generated from the 10,000 images for each digit. A good model should generate all digits uniformly covering all the distribution of the dataset. Figure 20 shows the counts of each modality

| Model | Avg Coherence | Avg FID |
|---|---|---|
| MVAE | $0.751_{(\pm 0.001)}$ | $139.47_{(\pm 0.055)}$ |
| MMVAE | $0.969_{(\pm 0.055)}$ | $176.49_{(\pm 0.135)}$ |
| MoPoE | $0.982_{(\pm 0.001)}$ | $163.31_{(\pm 0.590)}$ |
| MVTCAE | $0.983_{(\pm 0.001)}$ | $77.69_{(\pm 0.200)}$ |
| MMVAE+ | $0.937_{(\pm 0.130)}$ | $110.65_{(\pm 0.0005)}$ |
| SBM-VAE | $0.967_{(\pm 0.001)}$ | $73.29_{(\pm 0.080)}$ |
| SBM-VAE-C | $0.989_{(\pm 0.001)}$ | $72.96_{(\pm 0.06)}$ |
| SBM-VAE-T | $0.894_{(\pm 0.001)}$ | $77.33_{(\pm 0.05)}$ |
| SBM-RAE | $0.968_{(\pm 0.001)}$ | $81.70_{(\pm 0.120)}$ |
| SBM-RAE-C | $0.988_{(\pm 0.001)}$ | $83.52_{(\pm 0.10)}$ |
| SBM-RAE-T | $0.940_{(\pm 0.001)}$ | $86.48_{(\pm 0.09)}$ |

Table 5: Conditional Performance

| $\beta$ | Avg Coherence | Avg FID |
|---|---|---|
| 0.1 | $0.905_{(\pm 0.001)}$ | $52.12_{(\pm 0.045)}$ |
| 0.5 | $0.780_{(\pm 0.001)}$ | $102.85_{(\pm 0.050)}$ |
| 1 | $0.714_{(\pm 0.003)}$ | $108.35_{(\pm 0.295)}$ |

Table 6: Effect of $\beta$ on SBM-VAE

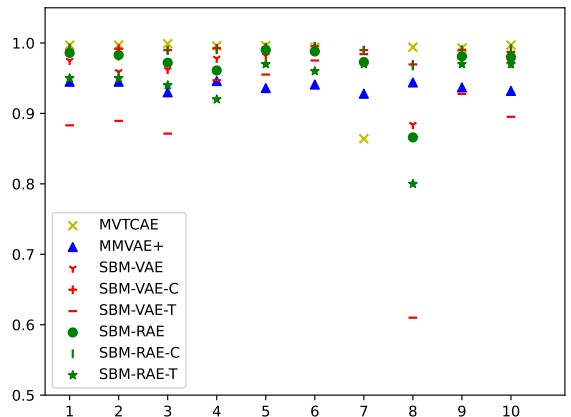

Figure 14: Conditional Accuracy (Coherence) of each modality given the rest.

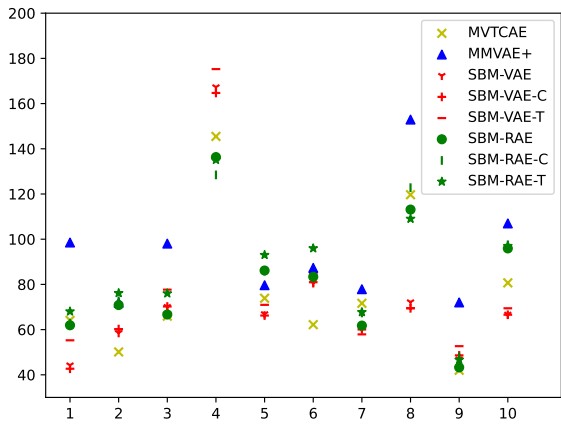

Figure 15: Conditional FID of each modality given the rest.

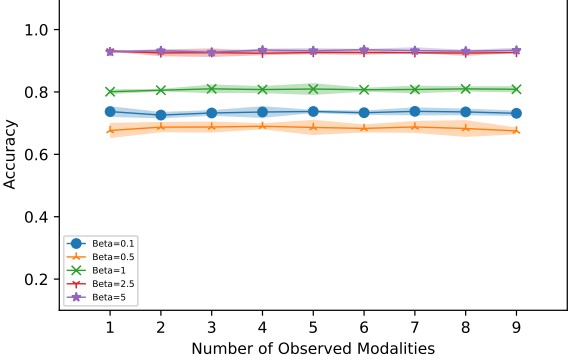

Figure 16: The conditional accuracy of the last modality generated by incrementing the given modality at a time for different $\beta$s of MMVAE+

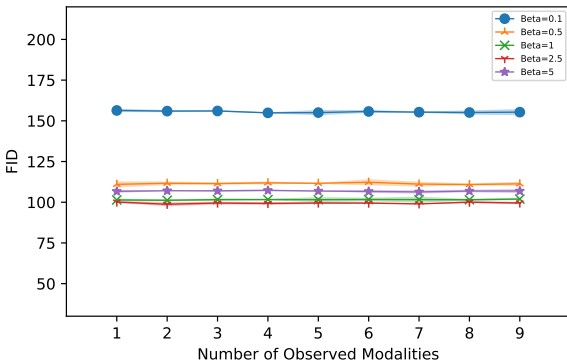

Figure 17: The conditional FID of the last modality generated by incrementing the given modality at a time for different $\beta$s of MMVAE+.

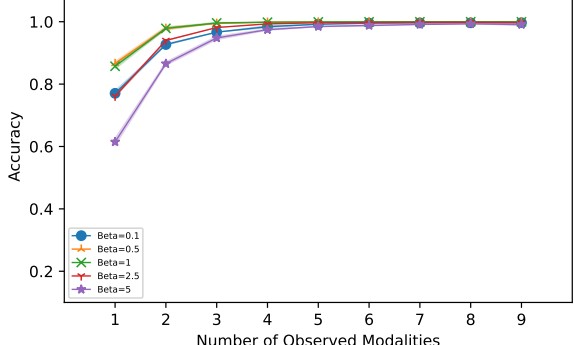

Figure 18: The conditional accuracy of the last modality generated by incrementing the given modality at a time for different $\beta$s of MVTCAE

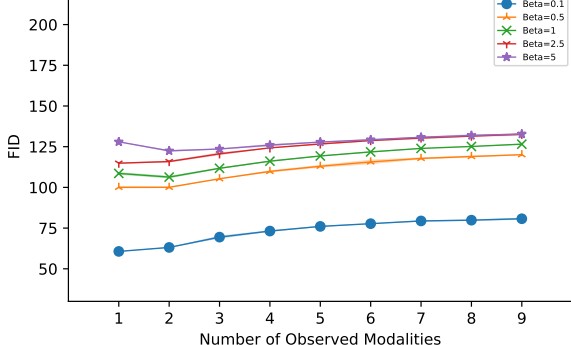

Figure 19: The conditional FID of the last modality generated by incrementing the given modality at a time for different $\beta$s of MVTCAE.

in a bar graph for SBMVAE, SBMRAE, MMVAE+, and MVTCAE. Our models and MMVAE+ cover most modes almost uniformly where as MVTCAE has some non-uniformity. Our models also have the highest

unconditional coherence which means they also generate coherent outputs at the same time which is an ideal property.

### A.5.1 Fine-tunning the generative models using missing modalites

We can further finetune the generative model (decoder) to increase the overall coherence. During training, we assume all the modalities are present. This condition is necessary for training $p_\theta$ in the described setup. However, we are interested in conditional queries of $p(.|\mathbf{x_o})$, we can achieve a tighter lower bound by further optimizing the $p_\psi(\mathbf{x_u}|\mathbf{z}_u)$ for sample from $q(\mathbf{z_u}|\mathbf{z_o}, \mathbf{x_o})$. We update the parameters of decoders to maximize the conditional log-likelihood:

$$\max_\psi \mathbb{E}_{q(\mathbf{z_u}|\mathbf{z_o},\mathbf{x_o})} \log p_\psi(\mathbf{x_u}|\mathbf{z_u})$$

$$= \max_\psi \frac{1}{K} \sum_{k=1}^{K} \log p_{\psi_k}(\mathbf{x_u^k}|\mathbf{z_u^k}) \quad z_\mathbf{u}^k \sim q(.|\mathbf{z_o}, \mathbf{x_o}) \tag{11}$$

For each training example in the batch, we randomly drop each modality with probability $p$. Eq. 11 will increase the likelihood of the true assignment of the dropped modalities given the observed modalities. We evaluate this experiments on a different score model trained using NCSN Song & Ermon (2019) and with unimodal VAEs with $\beta$=0.5.

Our experiments on fine-tuning show that the model's conditional performance increases very slightly, but the quality, measured by FID, drops significantly.

### A.5.2 Additional evaluation metrics

In this section, we want to include additional metric for the extended PolyMnist. In addition to using a classifier and comparing accuracy predicted using the classifier, here we will use cosine similarity and measure how latent variable $\mathbf{z}$ of each modality is recovered. We will the cosine similarity of each recovered $\mathbf{z}$ with true encoded $z$ for SBM-VAE-C. For comparison, we also include MVTCAE, which uses the product of experts. Table 7 shows the result of this section. The SBM model recovers the latent $\mathbf{z}$ that is conditionally generated and it has a higher cosine similarity with the ground truth encoded $\mathbf{z}$.

### A.5.3 Qualitative Results for Extended PolyMNIST

Here we show some conditionally generated samples and unconditional generation from each model. Conditional samples are shown in figures 24 and 25 and unconditional samples in figure 26.

| Modality | SBM-VAE cos sim | MVTCAE cos sim |
|---|---|---|
| 0 | 0.157 | 0.184 |
| 1 | 0.164 | 0.161 |
| 2 | 0.165 | 0.138 |
| 3 | 0.180 | 0.063 |
| 4 | 0.187 | 0.125 |
| 5 | 0.229 | 0.131 |
| 6 | 0.078 | 0.158 |
| 7 | 0.107 | 0.158 |
| 8 | 0.183 | 0.040 |
| 9 | 0.192 | 0.175 |
| Avg | **0.164** | 0.118 |

Table 7: Cosine similarity between the encoded $\mathbf{z}$ of the test data with the conditionally generated one given the rest for each modality

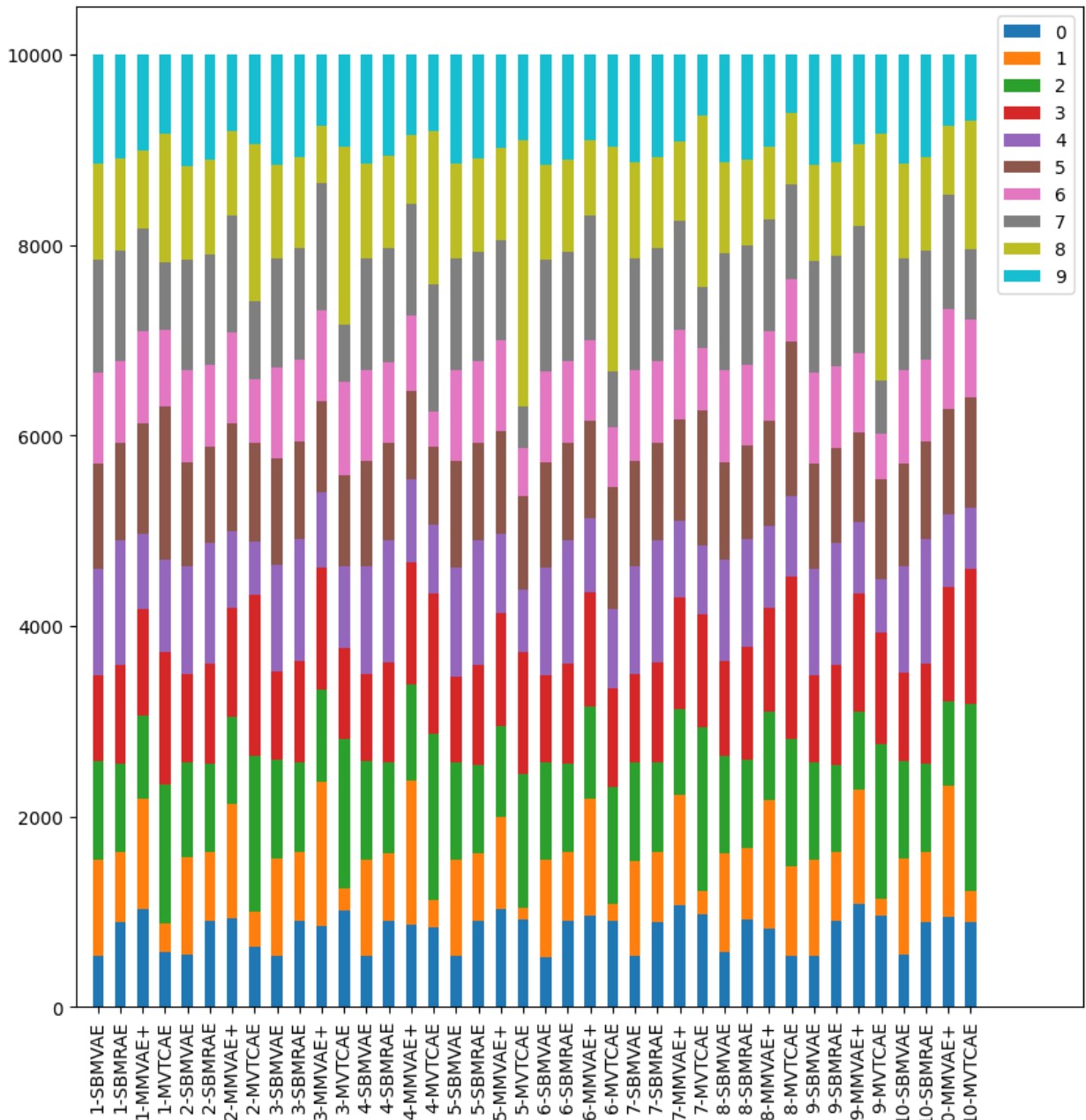

Figure 20: Mode Coverage

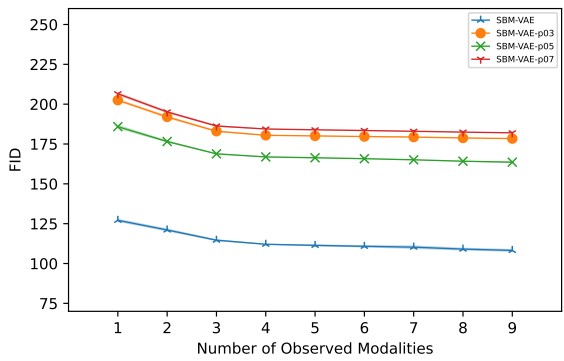

Figure 21: Plot of finetuning experiment using different $p$ values on how it affects conditional FID (different score model). The conditional FID of the last modality generated by incrementing the given modality at a time. The x-axis shows how many modalities are given to generate the modality and the y-axis shows the FID of the generated modality.

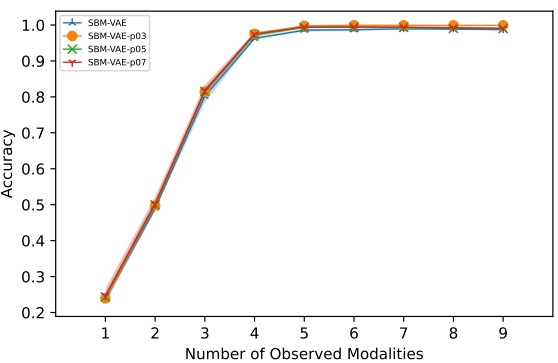

Figure 22: Plot of finetuning experiment using different $p$ values on how it affects conditional coherence (different score model). The conditional accuracy of the last modality generated by incrementing the given modality at a time. The x-axis shows how many modalities are given to generate the modality and the y-axis shows the accuracy of the generated modality.

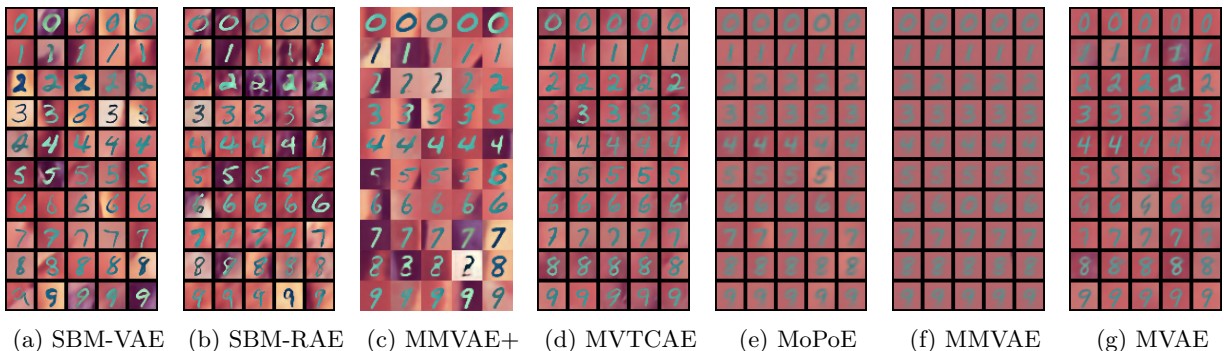

(a) SBM-VAE   (b) SBM-RAE   (c) MMVAE+   (d) MVTCAE   (e) MoPoE   (f) MMVAE   (g) MVAE

Figure 23: Conditional Samples from the third modality given the rest

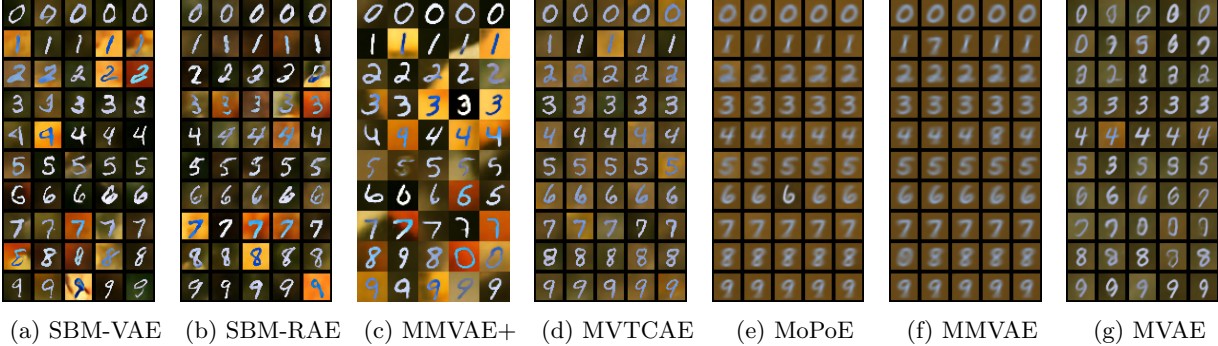

(a) SBM-VAE   (b) SBM-RAE   (c) MMVAE+   (d) MVTCAE   (e) MoPoE   (f) MMVAE   (g) MVAE

Figure 24: Conditional Samples from the first modality given the rest

## A.6 MHD Dataset

In this section, we discuss an additional experiment consisting of an audio modality. We use the audio and image modalities from the MHD dataset by Vasco et al. (2022). The image modality is from the MNIST

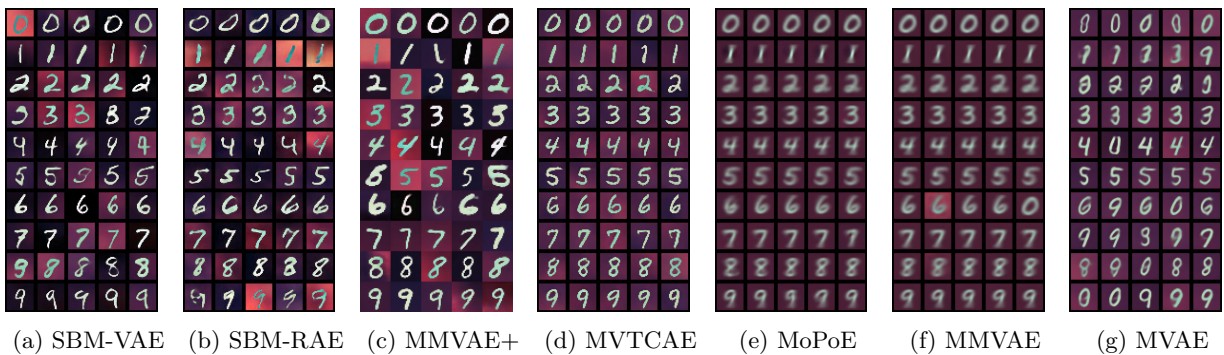

(a) SBM-VAE    (b) SBM-RAE    (c) MMVAE+    (d) MVTCAE    (e) MoPoE    (f) MMVAE    (g) MVAE

Figure 25: Conditional Samples from the sixth modality given the rest

(a) SBM-VAE

(b) SBM-RAE

(c) MMVAE+

(d) MVTCAE

(e) MoPoE

(f) MMVAE

(g) MVAE

Figure 26: Unconditional Samples where each of the columns are unconditional samples from each modality

dataset and the audio is a recording of each digit. We use the same encoder-decoder architecture from Vasco et al. (2022) with latent size 64. We conduct experiments on three models: SBMVAE, MVTCAE, and

MMVAE+ and we report the result in Table 8. The result shows our approach works in the presence of an audio modality.

Table 8: Generation Coherence on MHD Image-Audio

| Model | Audio→Image | Image→Audio | Unc |
|---|---|---|---|
| SBMVAE | $\mathbf{0.853}_{(\pm0.001)}$ | $\mathbf{0.861}_{(\pm0.004)}$ | $\mathbf{0.839}_{(\pm0.004)}$ |
| MVTCAE | $0.654_{(\pm0.004)}$ | $0.817_{(\pm0.004)}$ | $0.365_{(\pm0.002)}$ |
| MMVAE+ | $0.451_{(\pm0.004)}$ | $0.442_{(\pm0.002)}$ | $0.248_{(\pm0.003)}$ |

## A.7 Computational Efficiency

As discussed in the conclusion section as one limitation, SBM variant models take longer time during inference compared to the other baselines. To show how the models are different during inference in terms of computational efficiency, we evaluate the time it takes to generate conditional samples from batch of 256 extended PolyMnist data. We first start with a model trained on 2 modalities and increase the modalities the model is trained on to ten modalities. We compare MoPoE which uses a mixture of product of experts for inference and SBMVAE. We use A100 GPU for computing the time the models take. Figure 27 shows the time the models take in seconds. SBM-VAE takes almost a constant time for all modalities, while MoPoE starts increasing exponentially as the number of modalities increases. For the CelebMaskHQ dataset, a similar batch of 256 takes approximately 95 seconds for inference while MoPoE takes approximately 0.05 seconds. Here MoPoE has a clear edge as we use 1000 timesteps for this dataset on the score-based model. But one advantage of the score-based model is that the inference timesteps can be significantly decreased without compromising that much quality. If we use 100 timesteps, the time required will decrease 10 folds to around 9.5 seconds, but the performance drops very slightly. For 100 timesteps, the model retains the same F1 performance for the attribute and mask, and only the FID of the image modality drops by a few points.

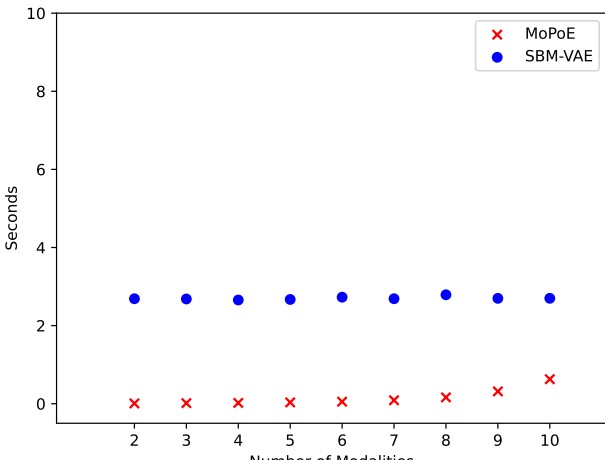

Figure 27: Time it takes for a batch of 256 on models trained from 2 modality to 10 modality

## A.8 Video-Audio Dataset

In this section, we want to show that our model can be scaled to higher dimensional datasets and modalities. We use partial samples, approximately 100K, from the soundnet dataset (Aytar et al., 2016) and train it on two modalities by separating the video and the audio as two modalities from each video. We follow the setup of CoDi (Tang et al., 2023) and use pre-trained image and audio auto-encoders. The audio is changed to melspectogram and encoded using an encoder to get $z_{\mathrm{aud}}$. We also use an autoencoder for the image and encode the image to $z_{\mathrm{img}}$ from a size of 256x256. To reconstruct the audio from the latent representation,

we first decode it and pass the audio melspectrogram to a Hifi-GAN model to generate audio samples. We train the joint SBM-VAE model using a UNET architecture from scratch on the combination of the two latent modalities, $z_{\text{aud}}$ and $z_{\text{img}}$. It's important to note that due to computation limitations, we only trained our model on the partial dataset, and generating good-quality video samples requires hundreds of millions of datasets. For reference, CoDi used approximately 112 Million dataset samples for video generation. In contrast, we are only using 100K samples of videos from soundnet and we split each one into a 2-second video with 8 frames per second and feed the corresponding audio as another modality. Figure 28, 29, 30, and 31 show the frames of the 2-second video generated and the audio can't be displayed here. Please look at the attached assets folder for more samples. Even though our model is not on par with current state-of-the-art video-audio generative models, partially because it's trained on a few datasets, it can be seen that it has the potential to be scaled to high-dimensional modalities. We have also added conditional and unconditional video and audio generation quantitative results in Table 9. We use FID to measure the quality of the generated image frames and the FAD metric using VGG features to evaluate the quality of the audio on 2000 held-out samples. CoDi$_1$ shows the result of the CoDi model on the same dataset. CoDi generates images and audios that are more general to this dataset, hence the lower result, but in general, the CoDi model performs much better, as shown in CoDi$_2$, which is evaluated on CoCo text-to-image FID and AudioCaps FAD where the results are taken from the paper Tang et al. (2023). In addition, CoDi can't perform unconditional generation, which is a disadvantage of the model. The unconditional results for this model are left empty because of that.

Table 9: Video generation Performance using SBM-VAE and CoDi

|  | SBMVAE | | CoDi$_1$ | | CoDi$_2$ | |
| --- | --- | --- | --- | --- | --- | --- |
|  | FID | FAD | FID | FAD | FID | FAD |
| Unconditional | 99.02 | 3.31 | - | - | - | - |
| Conditional | 101.19 | 8.23 | 100.4 | 14.9 | 11.26 | 1.80 |

## A.9 CelebMaskHQ Experimental Setup

The CelebMaskHQ dataset is taken from Lee et al. (2020) where the images, masks, and attributes are the three modalities. All face part masks were combined into a single black-and-white image except the skin mask. Out of the 40 attributes, 18 were taken from it similar to the setup of Wu & Goodman (2018). The encoder and decoder architectures are similar to Daniel & Tamar (2021) except MMVAE+ for the same reason as in PolyMNIST. A latent size of 256 was used for SBM models and MMVAE+, and a latent size of 1024 was used for MVTCAE and MoPoE. For MMVAE+, modality-specific and shared latent sizes are each 128 trained with IWAE estimator with K=1. For SBM-VAE, the image VAE was trained using Gaussian likelihood, posterior, and prior with $\beta = 0.1$. The same applies for the mask VAE but with $\beta = 1$. The attribute VAE uses Gaussian prior and posterior with a bernoulli likelihood. This applies to all other baselines with the exception of MMVAE+ which uses laplace likelihood and prior. We select the best $\beta$ for the baselines from [0.1,0.5,1,2.5,5]. MoPoE use $\beta$ of 0.1, MVTCAE 0.5 and MMVAE+ 5. For SBM-RAE, $\beta$ values of $10^{-4}$, $10^{-5}$, $10^{-4}$ and Gaussian noises of mean 0 and variance of 0.001, 0.001, 0.1 were added to $z$ before feeding to the decoder for the image, mask, and attribute modality respectively and the best performing one was selected.

The score-based models use a UNET architecture with the latent size reshaped into a size of 16x16. We take the mean of the posterior during training and inference time for SBM-VAE where as the **z** were taken during both times for SBM-RAE. Table 10 shows the detailed hyperparameters used for the score models. DiffuseVAE hyperparameters and models are the same ones used in Pandey et al. (2022) with formulation 1 for the 128x128 CelebHQ dataset. The energy-based model uses an MLP network. We train 3 pairs of models for each combination of modality. The supervised classifier for the attribute modality put for reference has a similar architecture to the encoder, and the features are projected for classification and trained with cross-entropy loss. The mask classifier uses a UNET architecture with an input dimension of 3 for the image and an output dimension of 1 to predict the masks.

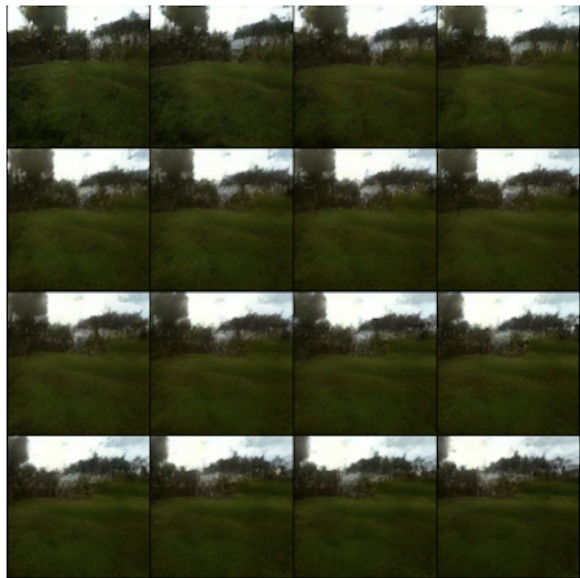

Figure 28: 16 video frames (ordered sequentially from left to right and top to bottom) generated unconditionally of a view of trees taken by someone moving along with audio modality (can't be shown here) of the background noise of a road. Please see the attachment for video samples.

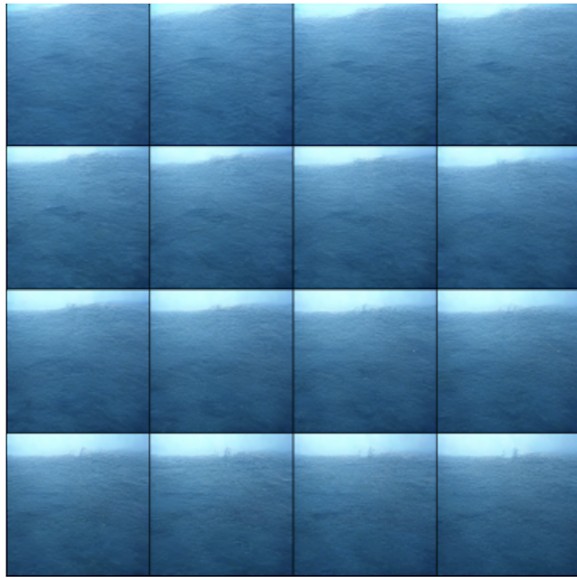

Figure 29: 16 video frames (ordered sequentially from left to right and top to bottom) generated unconditionally of a water wave with audio modality (can't be shown here) of a background noise of a water wave. Please see the attachment for video samples.

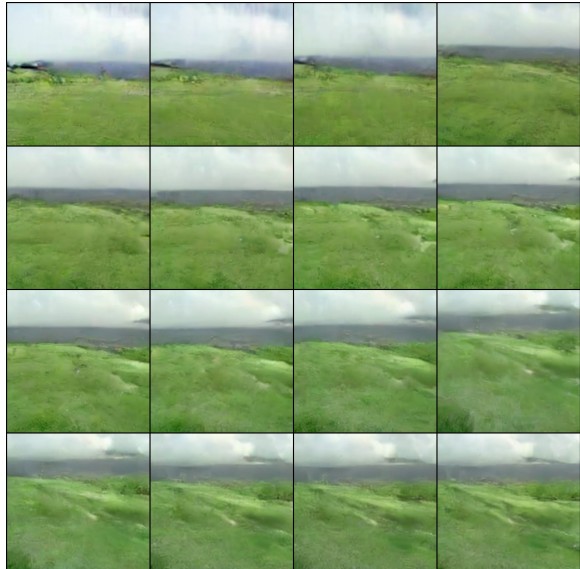

Figure 30: 16 video frames (ordered sequentially from left to right and top to bottom) generated unconditionally of a view of a road (audio can't be shown here). Please see the attachment for video samples.

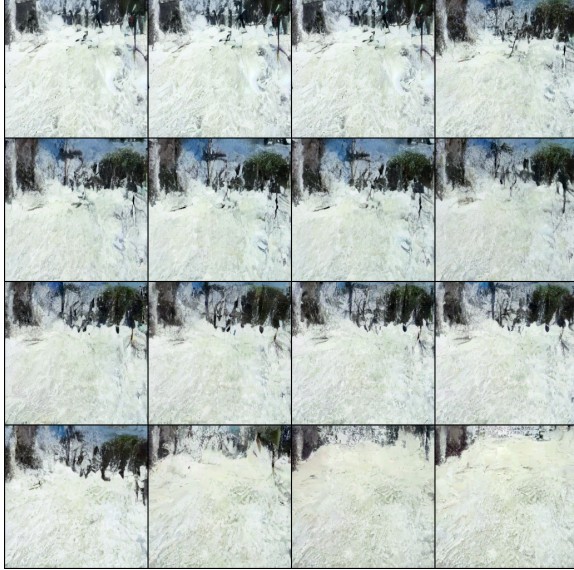

Figure 31: 16 video frames (ordered sequentially from left to right and top to bottom) generated unconditionally of a snowy area with trees (audio can't be shown here). Please see the attachment for video samples.

For the contrastive guidance case of SBM-VAE-T or SBM-RAE-T, we trained the auxiliary conditioning encoders using contrastive learning. Their architecture is similar to the encoder of the VAEs plus a linear project head that projects the modalities to a latent size of 512. We align the **z** representations of each

auxiliary encoder using contrastive learning objective so that the similarity between the **z**s coming from the same pair of multimodal data are maximized and the similarity of the pairs of **z**s coming from unrelated pairs are minimized. MLD by Bounoua et al. (2024) is trained using the same experimental setup used by the SBM models, the same train/val/test split, and the same architecture using the code provided here[3]

| Model | $\beta$min | $\beta$max | N | LD per step | EBM-scale | batch-size |
|-------|-----------|-----------|------|-------------|-----------|------------|
| SBM | 0.1 | 20 | 1000 | 1 | 2000 | 256 |

Table 10: Score Hyperparameters CelebMaskHQ

### A.10   CelebA Extended Experiments

### A.10.1   Training with missing modaliites

Multimodal models are trained on paired data coming from all modalities. Sometimes, some modality may be missing from a data sample. In those cases, it's important to make multimodal data trainable in these circumstances. We add this capability to the score-based model by using masking. Specifically, we mask out parts of the output score produced from the missing data. The missing data will be initially filled with noise. To simulate the performance of the model on missing training data, we take the CelebMaskHQ dataset and removing some portion of the data from the dataset. For example, we remove 20% of the data from each modality randomly and mix them. Note that multimodal data points that are not full are much larger than 20% as we first remove 20% from each modality randomly and then mix them. We evaluate the performance of the model as shown in Table 11. Overall, MVTCAE has good performance when training data is incomplete. SBMVAE has comparative outputs with the best Image FID.

| Model | Mask F1 | Attr F1 | Img FID |
|-------|---------|---------|---------|
| SBMVAE | $0.837_{(\pm 0.02)}$ | $0.627_{(\pm 0.01)}$ | $\mathbf{80.96}_{(\pm 0.2)}$ |
| MoPoE | $0.853_{(\pm 0.001)}$ | $0.680_{(\pm 0.02)}$ | $116.5_{(\pm 0.5)}$ |
| MVTCAE | $\mathbf{0.895}_{(\pm 0.04)}$ | $\mathbf{0.695}_{(\pm 0.03)}$ | $99.49_{(\pm 0.4)}$ |

Table 11: Missing training data performance evaluated given the other modalities on the whole test set

---

[3]Code for MLD https://openreview.net/forum?id=s25i99RTCg&noteId=bddHj6JHj1

### A.10.2   High Quality Image generation for CelebAHQ

In our setup, we use a normal variational autoencoder which makes it suitable for baseline comparison and training compute requirements. But since variational autoencoders suffer from low-quality and blurry images, compared to other SOTA generative models such as diffusion models (Dhariwal & Nichol, 2021), in order to get higher-quality, we can do two things. The first is to use a high-quality pre-trained auto-encoder and train the score model on it which is possible because we have independent two phase training stages. The second is to further increase the quality of the output of the decoder using DiffuseVAE model (Pandey et al., 2022). We illustrate the latter case here where the generated samples from SBM-VAE are fed into DiffuseVAE as shown in Figure 32. The DiffuseVAE helps in generating high-quality images from the low-quality ones without changing the image characteristics. As the figure shows, the quality of the images is much better while preserving the attributes and the masks given to generate them.

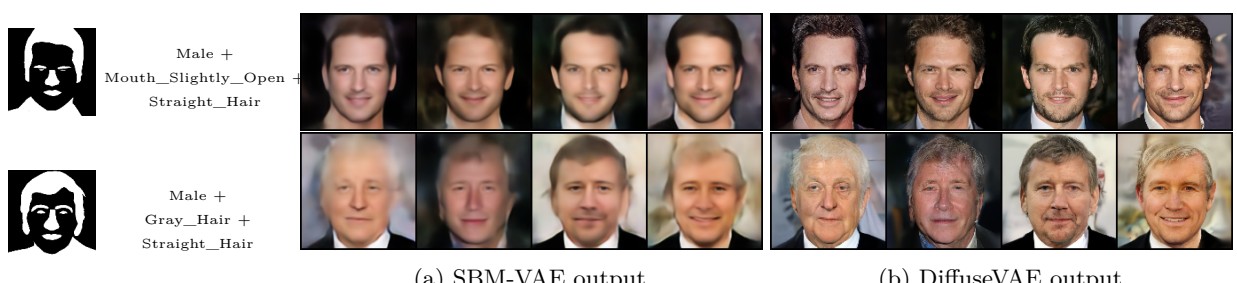

(a) SBM-VAE output                    (b) DiffuseVAE output

Figure 32: Higher quality image generation using DiffuseVAE given mask and attribute shown in the first two columns

### A.10.3   CelebMaskHQ Qualitative Result

In this section, we show samples from the CelebHQMASK dataset where the generated images are conditioned on different modalities. Figure 33 shows unconditionally generated outputs from each modality. Figure 34 shows different samples where only the image is given, Figure 35 shows different samples where the mask is given, Figure 37 shows different samples where the attribute is given.

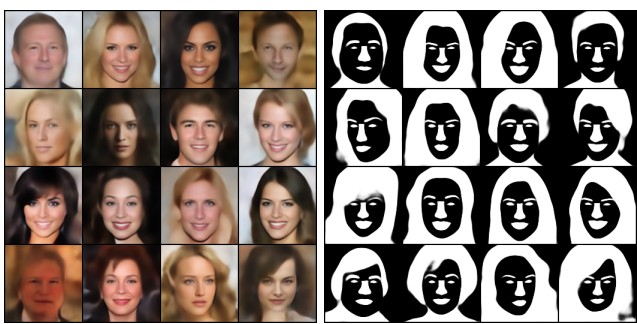

(a) SBM-RAE image       (b) SBM-RAE mask

(c) SBM-RAE attribute

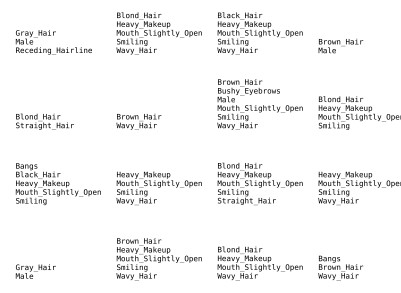

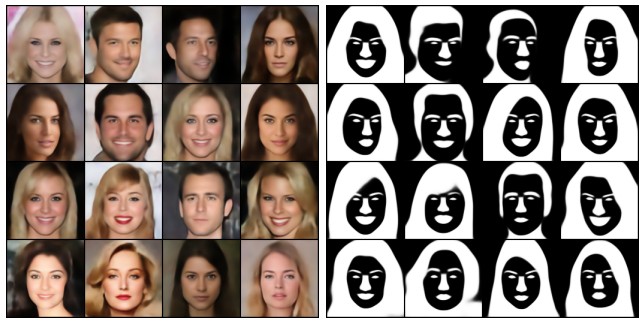

(d) SBM-VAE image       (e) SBM-VAE mask

(f) SBM-VAE attribute

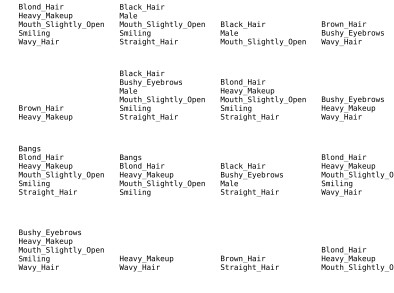

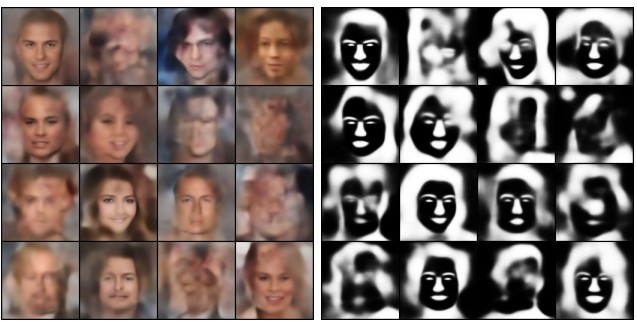

(g) MoPoE image       (h) MoPoE mask

(i) MoPoE attribute

Figure 33: Unconditional generation

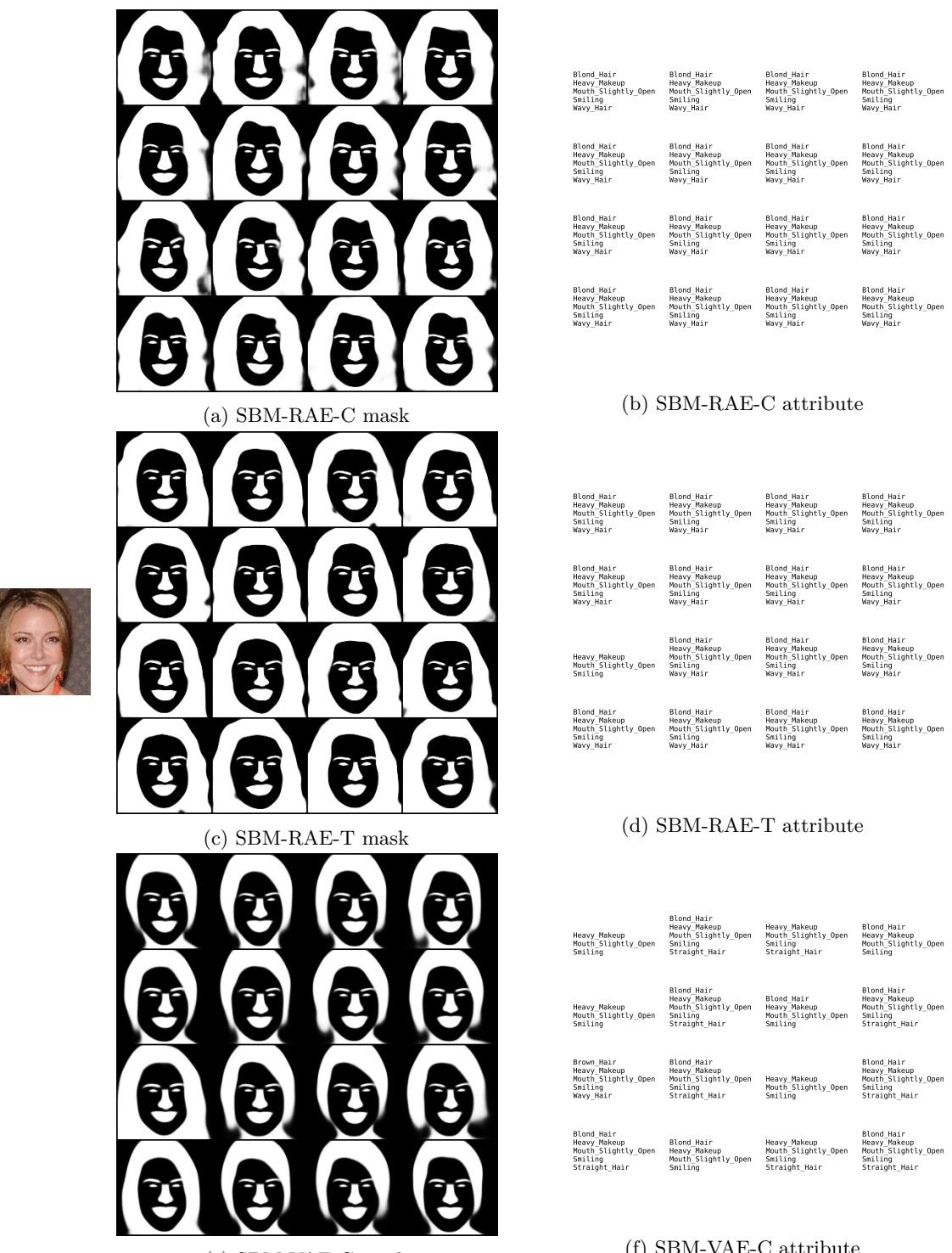

(a) SBM-RAE-C mask

(b) SBM-RAE-C attribute

(c) SBM-RAE-T mask

(d) SBM-RAE-T attribute

(e) SBM-VAE-C mask

(f) SBM-VAE-C attribute

Figure 34: Mask and Attribute generation given Image

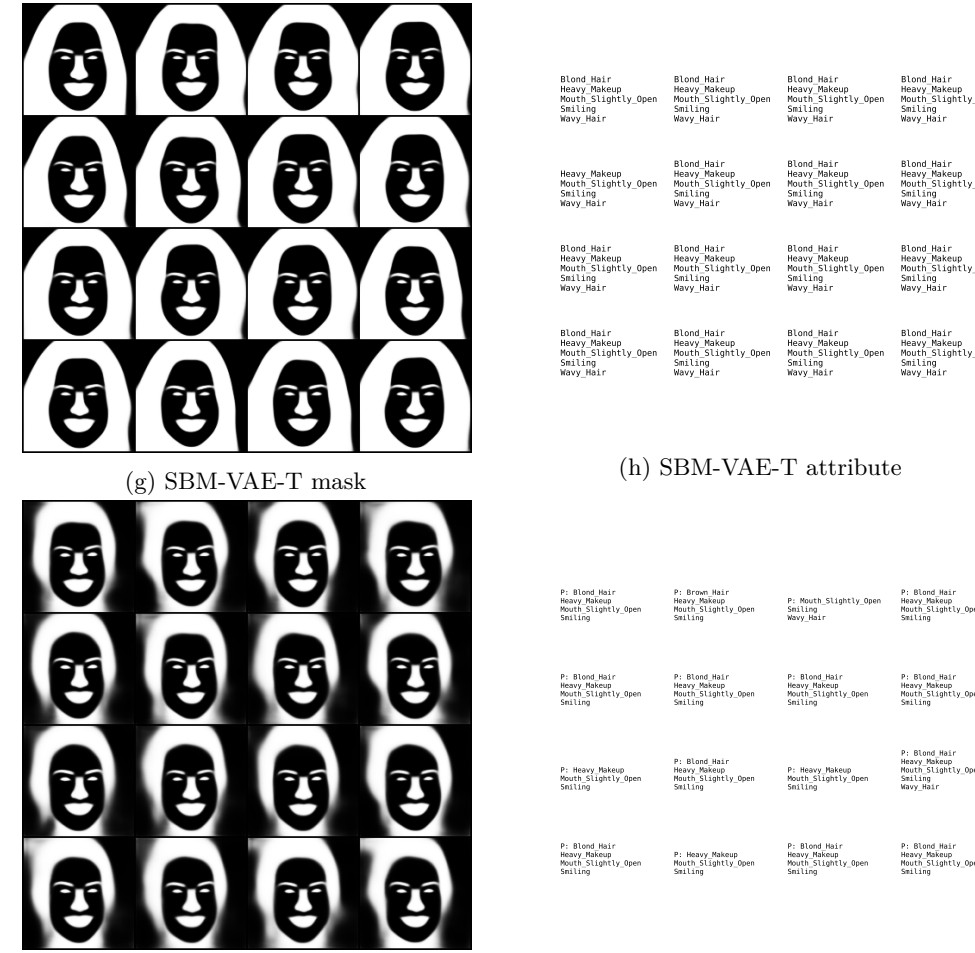

(g) SBM-VAE-T mask

(h) SBM-VAE-T attribute

(i) MoPoE mask

(j) MoPoE attribute

Figure 34: Mask and Attribute generation given Image

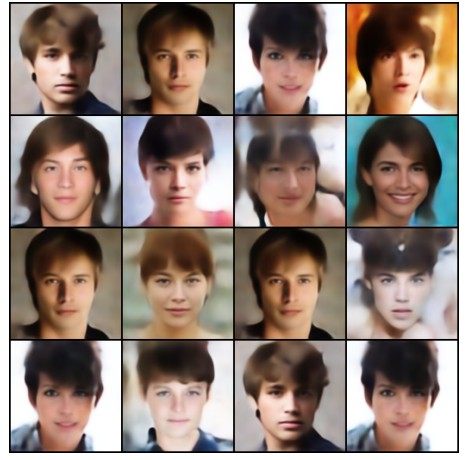

(a) SBM-RAE-C image

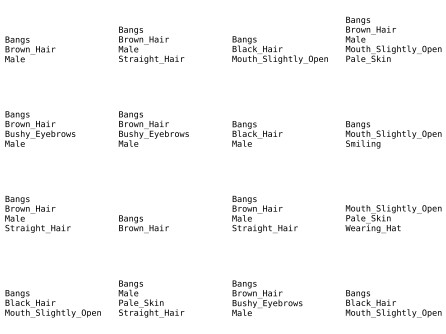

(b) SBM-RAE-C attribute

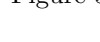

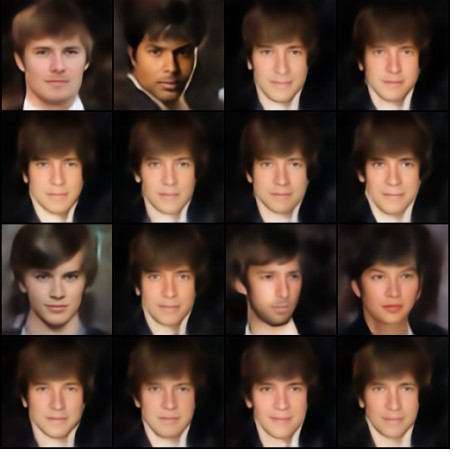

(c) SBM-RAE-T image

(d) SBM-RAE-T attribute

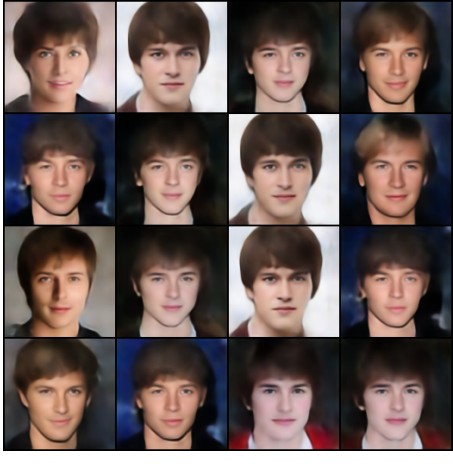

(e) SBM-VAE-C image

(f) SBM-VAE-C attribute

Figure 35: Image and Attribute generation given Mask

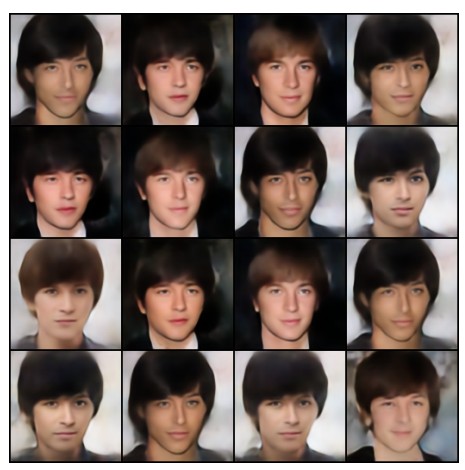

(g) SBM-VAE-T image

| | | | |
|---|---|---|---|
| Bangs
Black_Hair
Bushy_Eyebrows
Male | Bangs
Black_Hair
Male
Straight_Hair | Bangs
Brown_Hair
Male
Straight_Hair | Bangs
Black_Hair
Bushy_Eyebrows
Male |
| Bangs
Black_Hair
Male
Straight_Hair | Bangs
Brown_Hair
Male
Straight_Hair | Bangs
Black_Hair
Bushy_Eyebrows
Male
Mustache | Bangs
Black_Hair
Bushy_Eyebrows
Male
Straight_Hair |
| Bangs
Brown_Hair
Male
Wavy_Hair | Bangs
Black_Hair
Male
Straight_Hair | Bangs
Male
Straight_Hair | Bangs
Black_Hair
Bushy_Eyebrows
Male |
| Bangs
Black_Hair
Bushy_Eyebrows
Male | Bangs
Black_Hair
Bushy_Eyebrows
Male | Bangs
Black_Hair
Male | Bangs
Brown_Hair
Male |

(h) SBM-VAE-T attribute

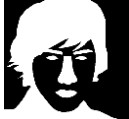

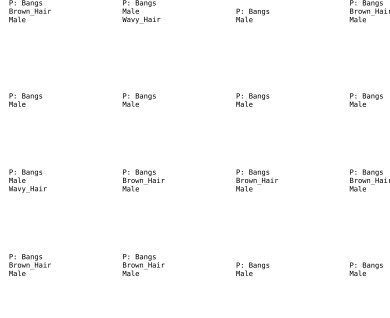

(i) MoPoE image

| | | | |
|---|---|---|---|
| P: Bangs
Brown_Hair
Male | P: Bangs
Male
Wavy_Hair | P: Bangs
Male | P: Bangs
Brown_Hair
Male |
| P: Bangs
Male | P: Bangs
Male | P: Bangs
Male | P: Bangs
Male |
| P: Bangs
Male
Wavy_Hair | P: Bangs
Brown_Hair
Male | P: Bangs
Brown_Hair
Male | P: Bangs
Brown_Hair
Male |
| P: Bangs
Brown_Hair
Male | P: Bangs
Brown_Hair
Male | P: Bangs
Male | P: Bangs
Male |

(j) MoPoE attribute

Figure 35: Image and Attribute generation given Mask

Black_Hair +
Bushy_Eyebrows +
Male +
Mustache +

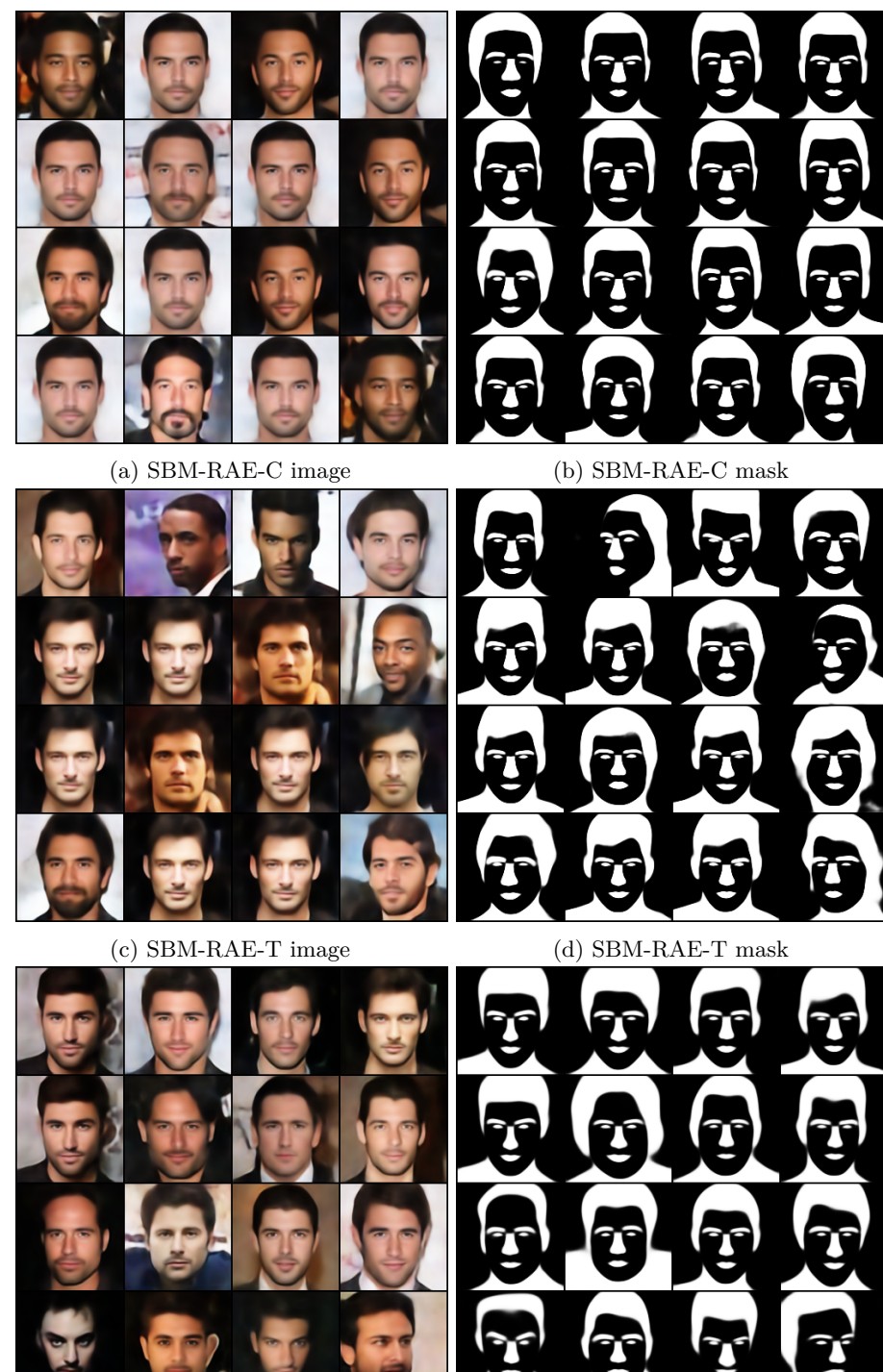

(a) SBM-RAE-C image       (b) SBM-RAE-C mask

(c) SBM-RAE-T image       (d) SBM-RAE-T mask

(e) SBM-VAE-C image       (f) SBM-VAE-C mask

Figure 36: Image and Mask generation given Attribute

Black_Hair +
Bushy_Eyebrows +
Male +
Mustache +

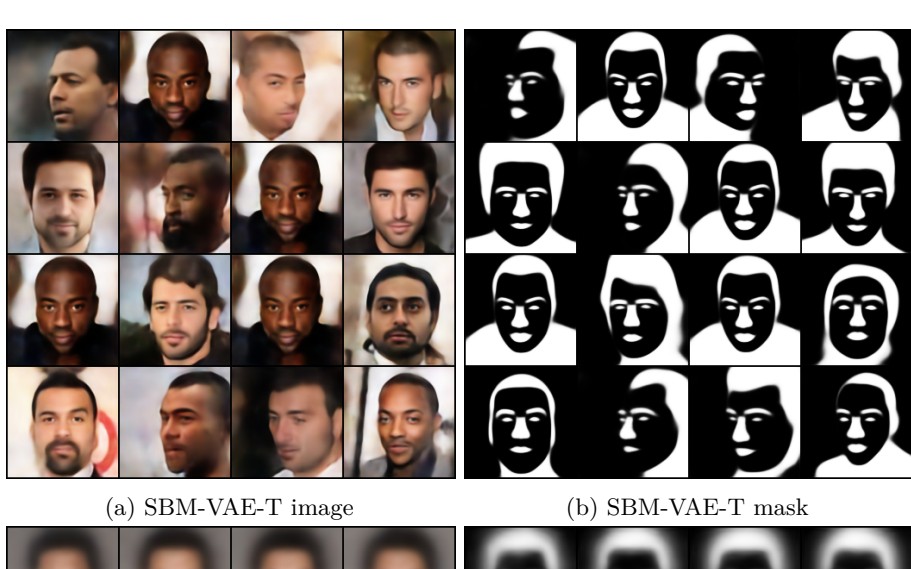

(a) SBM-VAE-T image      (b) SBM-VAE-T mask

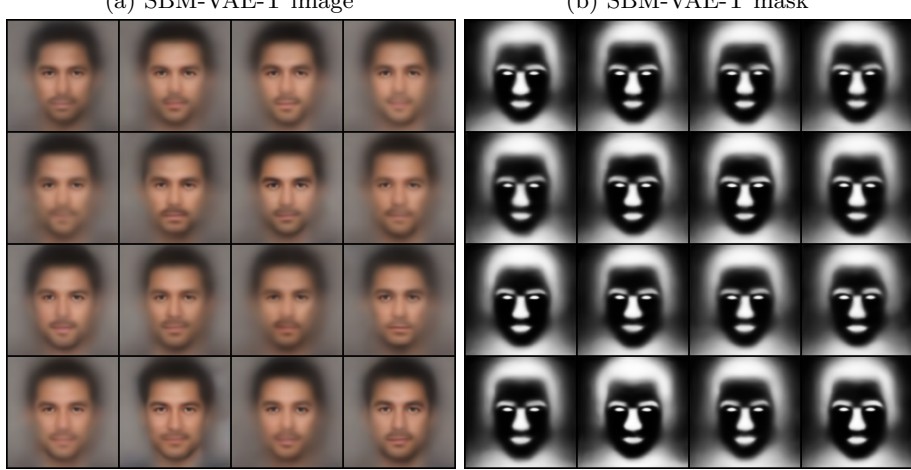

(c) MoPoE image      (d) MoPoE mask

Figure 37: Image and Mask generation given Attribute

