# OpenReview forum: "Score-Based Multimodal Autoencoder"
_TMLR — Accepted by TMLR_

### Review · Reviewer_aRPf · 2024-09-27

**Summary Of Contributions:**

Previous multimodal variational autoencoders (VAEs) models the joint posterior over the shared latent space with multimodal ELBO. This might result in generative discrepancy among modalities and no gain in additional conditional modalities. Instead, this paper proposes learning the correlation of individual latent spaces of unimodal representations using a score-based model (SBM). Furthermore, to increase coherence between the predicted modalities and the observed modalities, it provides coherence guidance by either using an energy-based model or aligning modalities through contrastive learning objectives. The resulting model achieves competitive or better performance compared to existing approaches on the the extended PolyMNIST and CelebAMask-HQ, in terms of both prediction coherence and generative quality. It is also more robust to adversarial attacks.

**Audience:**

Yes

**Broader Impact Concerns:**

Please include the broader impact statement in the final draft.

**Claims And Evidence:**

Yes

**Requested Changes:**

Please address the above concerns. Especially, it would be great to provide the results of training the proposed method on video data, using visual frames and corresponding audio signals.

**Strengths And Weaknesses:**

Strengths

- Overall, the methodology is well defined with the appropriate equations. Introducing the coherence guidance makes sense.
- Related works provide a thorough and detailed summary of previous research.
- The variants of proposed method achieve superior performance overall.

Weaknesses
- As the authors mentioned in Conclusion and Discussion, the proposed method is compute heavy. However, the authors did not compare computational efficiency between SBM variants and previous approaches, including inference time.
- I acknowledge that the experiments on PolyMNIST and CelebAMask-HQ prove the effectiveness of the proposed method. However, it makes the proposed method more practical if the method achieves superior performance on more high-dimensional data. SBM models are trained on images with a resolution up to 128x128 while vision community usually uses high-resolution images nowadays. Also, the connection between the audio and image modalities of the MHD dataset is quite unnatural. I don't think there is high correlation between written digits and the voice that reads it aloud.

Missing details
- The authors does not provide the details of the classifier to be used for measuring accuracy, including model architecture, training data, training recipes, etc.

---

> ### Author Response · Authors · 2024-10-28
> **Response from authors**
>
> Dear Reviewer,
>
> Thank you for your response. We will answer the questions that are raised here below.
> 1. “As the authors mentioned in Conclusion and Discussion, the proposed …”
>
> Thank you for raising this point. As noted in the Conclusion, SBM models require more computation time due to the iterative sampling process. To provide a concrete comparison, we measured the GPU compute time on the extended PolyMNIST dataset. SBMVAE takes approximately 2.7 seconds per batch of 256 samples, while MoPoE takes around 0.6 seconds for the same batch size. We have included these details in a new section in Appendix A.7 for further clarity.
>
>
> 2. “I acknowledge that the experiments on PolyMNIST and CelebAMask-HQ … “
>
> Thank you for this observation. It’s true that models trained on high-dimensional datasets typically require substantial compute resources and vast amounts of training data to achieve high-quality generation. For instance, generative models that produce state-of-the-art results often rely on datasets of considerable scale, such as the LAION-400M [1] and LAION-5B [2] datasets, which contain hundreds of millions to billions of samples. Scaling up to higher resolutions or additional modalities like video and audio further increases computational demands.
>
> Despite this, we have added a new model on video and audio modality trained using around 100,000 partial samples from the Soundnet dataset. We clipped the videos to 2-second videos with 8 frames per second of 256x256 images and the corresponding audio as another modality. For reference, a diffusion generative model from CoDi [3] uses all of the soundnet, audioset, and other video datasets, approximately 112.6 million samples, to generate good samples. Compared to that, we used less than 1% of the video data because of the vast computation resources it takes and trained on partial soundnet data. We have included this experiment's result and discussion section in Appendix A.8. Additionally, we have included a variety of generated samples in the supplementary assets folder, as video frames and audio are challenging to convey fully within the main text. We hope this addition demonstrates the model’s flexibility, even with limited computing resources, while highlighting practical considerations in scaling to more complex data types.
>
>
> 3. “The authors does not provide the details of the classifier to be used … “
> Thank you for pointing that out. We have added the details of the classifier in Appendix A.2.
>
> [1] Christoph Schuhmann, Richard Vencu, Romain Beaumont, Robert Kaczmarczyk, Clayton Mullis, Aarush Katta, Theo Coombes, Jenia Jitsev, Aran Komatsuzaki:
> LAION-400M: Open Dataset of CLIP-Filtered 400 Million Image-Text Pairs.
>
> [2] Christoph Schuhmann, Romain Beaumont, Richard Vencu, Cade Gordon, Ross Wightman, Mehdi Cherti, Theo Coombes, Aarush Katta, Clayton Mullis, Mitchell Wortsman, Patrick Schramowski, Srivatsa Kundurthy, Katherine Crowson, Ludwig Schmidt, Robert Kaczmarczyk, Jenia Jitsev: LAION-5B: An open large-scale dataset for training next generation image-text models. NeurIPS 2022
>
> [3] Zineng Tang, Ziyi Yang, Chenguang Zhu, Michael Zeng, and Mohit Bansal. Any-to-any generation via composable diffusion. CoRR, abs/2305.11846, 2023. doi: 10.48550/ARXIV.2305.11846.

---

### Review · Reviewer_dCDR · 2024-10-02

**Summary Of Contributions:**

This paper proposes a novel approach to learning multimodal VAEs by first training individual unimodal VAEs for each modality and then modeling the intra-modality dependencies (i.e., the joint latent distribution) using score matching approaches. Their experiment results on extended PolyMNIST and CelebAMask-HQ have demonstrated a series of desired properties of a multi-VAE model.

**Audience:**

Yes

**Broader Impact Concerns:**

There are no significant ethical concerns from my point of view

**Claims And Evidence:**

Yes

**Requested Changes:**

1. In the first row of page 4, is there a particular reason that the score is defined as an approximation of the log derivative of the distribution instead of just equality? One short sentence to clarify would be appreciated.

2. The latent coherence is evaluated solely based on label prediction in a discrete space. It would be great to have an additional continuous measure for the latent coherence. For this, one could design a synthetic numerical experiment by sampling a set of correlated modality-specific latents and generating the corresponding observations that form different modalities.  After training, one can compare the correlation of the predicted latents and the ground truth ones to see how well the proposed model preserves the latent coherence.

3. Prior works have studied provable latent variable identification under the multi-modality setting (for example, [1, 2]). Identified latents have shown a series of desired properties that align with the expected properties discussed in this work. Is there any identifiability guarantee in this proposed approach? In general, I think it would be beneficial to include a discussion on (1) the identifiability of the proposed approach and (2) a connection with other identifiable frameworks under a multimodal setting.


**Reference**

[1] Imant Daunhawer, Thomas M. Sutter, Kieran Chin-Cheong, Emanuele Palumbo, and Julia E Vogt. On the limitations of multimodal VAEs. In International Conference on Learning Representations, 2022. URL https://openreview.net/forum?id=w-CPUXXrAj.

[2] Dingling Yao, Danru Xu, Sebastien Lachapelle, Sara Magliacane, Perouz Taslakian, Georg Martius, Julius von Kügelgen, and Francesco Locatello. Multi-view causal representation learning with partial observability. In The Twelfth International Conference on Learning Representations, 2024. URL https://openreview.net/forum?id=OGtnhKQJms.

**Strengths And Weaknesses:**

**Strengths**

This paper tackles the challenge of multimodal training from a novel and interesting perspective. Instead of directly learning the joint posterior, which becomes easily intractable in high dimensions, this approach smartly divides the problem into two sub-problems and solves them one by one in two different stages. It also avoids modality sub-sampling, which fails to provide modality coherence, as shown by [1].

**Weaknesses**

While this paper provides a practical algorithm demonstrating promising performance on the considered datasets, some theoretical assumptions seem incoherent with their use cases. For example, at the beginning of section 2, it is assumed that the latent variables only consist of modality-specific latent variables in the sense that none of them are shared between different modalities. On the other hand, this paper has proposed using contrastive guidance for score matching by incorporating an aligned representation learned using a shared encoder (section 2.2.2).  If, as stated in Section 2, no latents are shared between modalities, which kind of representation do we expect from this secondary contrastive encoder? And why would it then help guide score matching?

**Reference**

[1] Imant Daunhawer, Thomas M. Sutter, Kieran Chin-Cheong, Emanuele Palumbo, and Julia E Vogt. On the limitations of multimodal VAEs. In International Conference on Learning Representations, 2022. URL https://openreview.net/forum?id=w-CPUXXrAj.

---

> ### Author Response · Authors · 2024-10-28
> **Response from Authors**
>
> Dear Reviewer,
>
> Thank you for your response. We will answer the questions that are raised here below.
>
> 1. “While this paper provides a practical algorithm demonstrating promising performance …”
>
> Thank you for raising this important point regarding the model's theoretical assumptions and practical applications. Our assumption, as stated in Section 2, is that the modalities are conditionally independent given the latent representations. This simplifies the joint generative model $p(x_{1:M} | z_{1:M})$  and the joint recognition model $q(z_{1:M} | x_{1:M})$, enabling us to factorize the distributions and setup the theoretical formulation. However, as you pointed out, this independence assumption may not hold perfectly in practice, especially given the complex interdependencies in real-world multimodal data and the capability of the score-based model to capture it. To address this, we incorporated contrastive guidance for score matching by aligning representations across modalities using a shared encoder. This secondary contrastive encoder projects each modality into a shared latent space, compensating for dependencies that the score-based model alone may not fully capture. By doing so, the contrastive guidance improves model robustness, particularly when the number of observed modalities is low, as it encourages the model to be conditioned on aligned latent representations more effectively across modalities. The guidance of this method is also not directly used to generate the other modalities like other mixture-based multimodal VAE generative models. This means it will not be a limitation factor and won’t degrade the generative quality of the outputs; it only acts to provide guidance information towards the correct output.
>
> In essence, while the independence assumption helps simplify our generative and recognition models theoretically, the contrastive guidance provides a practical solution to reinforce performance, especially when one or two modalities are observed. We appreciate your insightful question and hope this answer clarifies your question.
>
> 2. “In the first row of page 4, is there a particular reason that the score is defined as a …”
>
> Thank you for highlighting this. The approximation symbol is used to indicate that the neural network does not output the exact score value of  $\nabla_{\mathbf{z}} \log(p(\mathbf{z}))$, but rather approximates it based on the neural network’s parameters [1]. To avoid confusion with  $\theta$ appearing on both sides, we have removed  \theta from the right side to make this distinction clearer.
>
> 3. “The latent coherence is evaluated solely based on label prediction in a discrete space …”
>
> Thank you for the suggestion. In addition to the accuracy-based evaluations, we have incorporated a continuous metric to assess latent coherence based on your recommendation. Specifically, we have added a new experiment in Appendix A.5.2 that uses cosine similarity as a continuous metric to measure alignment between the ground-truth encoded z  and the model’s recovered z, conditionally generated from the other modalities on the extended PolyMNIST dataset.
>
>
> 4. “Prior works have studied provable latent variable identification …”
>
> Thank you for the question. Identifiability in multimodal settings generally focuses on recovering a shared latent source from multiple data views [2]. Works related to identifiability assume recovering a shared source up to some noise that generated an outcome. In contrast, our work assumes the latent $\textbf{z}$ of individual VAEs holds all the information about each modality, and we are concerned with recovering the $\textbf{z}$ specific to each modality instead of a shared source $\textbf{z}$ that generated the outputs. Our model differs by using joint modeling with score-based methods to maintain coherence across all modalities and learn the joint distribution, avoiding the need for constructing a shared latent variable that identifiability models rely on. However, we can interpret the joint latent space across all modalities as a “source” that generates multimodal outputs. Extending our work in this direction could involve evaluating block-identifiability by examining how well the joint latent representation captures shared factors among modalities. The joint latent representation is recovered using the score-based model, and the representation can then be used to predict ground truth factors. Here, we provide a simple theoretical guarantee that our method recovers the latent representation.
>
> continued ...

---

> > ### Author Response · Authors · 2024-10-28
> > **Continued from previous part**
> >
> > Since we assume latent variables only capture modality-specific representation and we don't have any shared representation, the identifiability problem reduces to recovering $z_i$ from $z_{-i}$.
> >
> > Assume we have only two modalities $X_1$ and $X_2$, and $z_1$ and $z_2$ are the corresponding modality-specific latent variables for each modality. For simplicity, assume that the dependence of $z_1$ and $z_2$ has been captured via a bivariate normal distribution. We will demonstrate how to identify $z_1$ given $z_2$ by utilizing the score function $ \nabla \log p(z_1, z_2) $ derived from the simple bivariate normal distribution.
> >
> > Consider two latent random variables $z_1$ and $z_2$ following a bivariate normal distribution with zero means, unit variances, and correlation coefficient $\rho$:
> >
> > $$
> > \begin{pmatrix}
> > z_1 \\\\
> > z_2
> > \end{pmatrix}
> > \sim \mathcal{N}
> > \left(
> > \begin{pmatrix}
> > 0 \\\\
> > 0
> > \end{pmatrix},
> > \begin{pmatrix}
> > 1 & \rho \\\\
> > \rho & 1
> > \end{pmatrix}
> > \right).
> > $$
> >
> > The joint probability density function (pdf) is:
> >
> > $$
> > p(z_1, z_2) = \frac{1}{2\pi \sqrt{1 - \rho^2}} \exp\left( -\frac{1}{2(1 - \rho^2)} \left( z_1^2 - 2 \rho z_1 z_2 + z_2^2 \right) \right).
> > $$
> >
> > The score function is the gradient of the log of the joint pdf with respect to $z_1$ and $z_2$:
> >
> > $$
> > \nabla_{\mathbf{z}} \log p(z_1, z_2) =
> > \begin{pmatrix}
> > \frac{\partial}{\partial z_1} \log p(z_1, z_2) \\\\
> > \frac{\partial}{\partial z_2} \log p(z_1, z_2)
> > \end{pmatrix}.
> > $$
> >
> > First, we express the log of the joint pdf:
> >
> > $$
> > \log p(z_1, z_2) = -\log(2\pi \sqrt{1 - \rho^2}) - \frac{1}{2(1 - \rho^2)} \left( z_1^2 - 2 \rho z_1 z_2 + z_2^2 \right).
> > $$
> >
> > Then we differentiate with respect to $z_1$:
> >
> > $$
> > \begin{aligned}
> > \frac{\partial}{\partial z_1} \log p(z_1, z_2) &= -\frac{1}{2(1 - \rho^2)} \frac{\partial}{\partial z_1} \left( z_1^2 - 2 \rho z_1 z_2 + z_2^2 \right) \\\\
> > &= -\frac{1}{2(1 - \rho^2)} \left( 2 z_1 - 2 \rho z_2 \right) \\\\
> > &= -\frac{1}{1 - \rho^2} \left( z_1 - \rho z_2 \right).
> > \end{aligned}
> > $$
> >
> > Similarly,
> >
> > $$
> > \frac{\partial}{\partial z_2} \log p(z_1, z_2) = -\frac{1}{1 - \rho^2} \left( z_2 - \rho z_1 \right).
> > $$
> >
> > Therefore, the score function is:
> >
> > $$
> > \nabla_{\mathbf{z}} \log p(z_1, z_2) = -\frac{1}{1 - \rho^2}
> > \begin{pmatrix}
> > z_1 - \rho z_2 \\\\
> > z_2 - \rho z_1
> > \end{pmatrix}.
> > $$
> >
> > Our goal is to find $z_1$ that maximizes the conditional probability $p(z_1 \mid z_2)$. Since $\log p(z_1 \mid z_2) = \log p(z_1, z_2) - \log p(z_2)$, and $\log p(z_2)$ is constant with respect to $z_1$, maximizing $\log p(z_1 \mid z_2)$ is equivalent to maximizing $\log p(z_1, z_2)$ with respect to $z_1$.
> >
> > We use gradient ascent to iteratively update $z_1$:
> >
> > $$
> > z_1^{(k+1)} = z_1^{(k)} + \alpha \frac{\partial}{\partial z_1} \log p(z_1^{(k)}, z_2),
> > $$
> >
> > where $\alpha > 0$ is the learning rate.
> >
> > Substitute the expression for the derivative:
> >
> > $$
> > \begin{aligned}
> > z_1^{(k+1)} &= z_1^{(k)} + \alpha \left( -\frac{1}{1 - \rho^2} \left( z_1^{(k)} - \rho z_2 \right) \right) \\\\
> > &= z_1^{(k)} - \frac{\alpha}{1 - \rho^2} \left( z_1^{(k)} - \rho z_2 \right).
> > \end{aligned}
> > $$
> >
> > Let $\lambda = \frac{\alpha}{1 - \rho^2}$:
> >
> > $$
> > z_1^{(k+1)} = z_1^{(k)} - \lambda \left( z_1^{(k)} - \rho z_2 \right) = z_1^{(k)} (1 - \lambda) + \lambda \rho z_2.
> > $$
> >
> > At convergence, $z_1^{(k+1)} = z_1^{(k)} = z_1^\ast$:
> >
> > $$
> > \begin{aligned}
> > z_1^\ast &= z_1^\ast (1 - \lambda) + \lambda \rho z_2 \\\\
> > \implies z_1^\ast &= \frac{\lambda \rho z_2}{\lambda} \\\\
> > \implies z_1^\ast &= \rho z_2.
> > \end{aligned}
> > $$
> >
> > Thus, the iteration converges to $z_1^\ast = \rho z_2$.
> >
> > For a bivariate normal distribution, the conditional distribution $z_1 \mid z_2$ is:
> >
> > $$
> > z_1 \mid z_2 \sim \mathcal{N} \left( \mu_{z_1 \mid z_2}, \sigma^2_{z_1 \mid z_2} \right),
> > $$
> >
> > where:
> >
> > $$
> > \mu_{z_1 \mid z_2} = \rho z_2, \quad \sigma^2_{z_1 \mid z_2} = 1 - \rho^2.
> > $$
> >
> > Our iterative method correctly identifies the conditional mean $z_1 = \rho z_2$.
> >
> > We can generalize this approach to arbitrary modalities and more complex models, given that we have enough data to train the score function accurately.
> >
> > [1] Yang Song and Stefano Ermon. Generative modeling by estimating gradients of the data distribution. Advances in Neural Information Processing Systems 32: Annual Conference on Neural Information Processing Systems 2019, NeurIPS 2019, December 8-14, 2019, Vancouver, BC, Canada, pp. 11895–11907, 2019
> >
> > [2] Luigi Gresele, Paul K. Rubenstein, Arash Mehrjou, Francesco Locatello, and Bernhard Schölkopf. The incomplete rosetta stone problem: Identifiability results for multi-view nonlinear ICA. Proceedings of the Thirty-Fifth Conference on Uncertainty in Artificial Intelligence,
> > UAI 2019, Tel Aviv, Israel, July 22-25, 2019, volume 115 of Proceedings of Machine Learning Research, pp.217–227. AUAI Press, 2019.

---

### Review · Reviewer_Rnrd · 2024-10-19

**Summary Of Contributions:**

The authors propose a way to fuse together a set of unimodal autoencoders into a multimodal autoencoder, using score matching to train the joint prior over the latent variables for all modalities. They use contrastive learning to classify when a pair of latent variables from two modalities correspond to the same example, and use this to improve the coherence across modalities. Their algorithm is empirically validated using two toy datasets (MNIST-based), and one realistic dataset (CelebAMask-HQ), where they compare against various multimodal VAE baselines.

**Audience:**

Yes

**Claims And Evidence:**

Yes

**Requested Changes:**

You might be able to improve the coherence guidance by using multiple positive pairs, as done in supervised contrastive learning (Khosla 2020). Given an anchor modality, the positive pairs are all other modalities from the same example, while the negative pairs are all other modalities from different examples.

I would be more in favor of acceptance if this paper included an additional realistic dataset such as VQAv2.

**Strengths And Weaknesses:**

The strength of this work is that the authors' algorithm is simple and scales well to a large number of modalities. We know how to train unimodal autoencoders well, and training the joint prior is a relatively simple generative modeling problem with low-dimensional (compared to the original inputs) observations. The coherence guidance is also a good idea, although it can likely be improved (see requested changes.)

The weakness of this work are the empirical results. The authors experiment with two toy MNIST-based problems (one in the appendix), and one realistic dataset. The paper would be stronger with an additional realistic dataset, as is fairly standard with recent empirically-oriented ML papers. Also, this is highly subjective, but the datasets that the authors work with are not canonical examples of multimodal learning. This is especially the case with PolyMNIST, where the "modalities" are what people typically call "style." I argue this is also the case for the image and mask in CelebAMask-HQ. I believe the paper would be much stronger with an additional dataset such as VQAv2.

---

> ### Author Response · Authors · 2024-10-28
> **Response from Authors**
>
> Dear Reviewer,
>
> Thank you for your response. We will answer the questions that are raised here below.
>
> 1. “The weakness of this work are the empirical results …”
>
> Thank you for your valuable feedback regarding the dataset selection and the empirical scope of our study. We appreciate your concern and have made revisions to strengthen this aspect of our work. While the MNIST-based datasets may not be high-dimensional, they are widely used in high-quality multimodal learning research as a baseline [1,2], allowing for direct comparisons and thorough benchmarking. We also chose PolyMNIST mainly due to its flexibility in scaling the modality count. This enables a controlled analysis of how our model performs in various multimodal configurations, which is fundamental to understanding its adaptability and robustness in different scenarios. As you have also mentioned, we have employed the CelebAMask-HQ dataset, which is more realistic and complex. We believe that both extended PolyMNIST and CelebAMask-HQ provide important insights.
>
> Regarding the VQAv2 dataset, we agree that it represents an important benchmark in multimodal learning, particularly for language-vision models and foundational large language models. The state-of-the-art works that use this dataset are answer generation to a question conditioned on an image [4]. However, it has been infrequently applied in the multimodal VAE or diffusion-based generative models, which generally focus on cross-modal coherence rather than language-driven interaction. In response to your suggestion for additional realistic data, we expanded our experiments by incorporating a video-audio setup using the SoundNet dataset [3]. This addition presents a unique, dynamic form of multimodal data, further testing the model’s capabilities and generalizability beyond the modalities we added. For clarity, we have detailed the experimental setup in Appendix A.8 of the revised paper, including a comprehensive description of the training process. Additionally, we have included a variety of generated samples in the supplementary assets folder, as video frames and audio are challenging to convey fully within the main text. We hope these additions and clarifications address your concerns.
>
>
> 2. “You might be able to improve the coherence guidance by using multiple positive pairs …”
>
> Thank you for this suggestion. We implemented a supervised contrastive learning approach for the coherence guidance, incorporating labels to leverage additional positive samples from the same class in the extended PolyMNIST dataset. This allowed us to treat other modalities of the same class as positive pairs and modalities from different classes as negatives, following the setup you proposed. In testing this approach, however, we observed only a very negligible change in the performance of the score-based model. To investigate that, we evaluated the auxiliary encoder’s baseline cosine similarity between aligned z representations of positive pairs from the validation set. Using our prior unsupervised approach, it was already high at 0.978. With the added label-based contrastive adjustment, this similarity rose to 0.998, indicating a strong alignment even before supervised contrastive learning was applied. This small improvement in cosine similarity corresponded with negligible performance gains in downstream tasks and therefore we didn’t change the result in the main paper. These results suggest that the initial coherence guidance without label information was already achieving sufficient alignment for this dataset. Nonetheless, we appreciate this valuable suggestion.
>
> [1] Emanuele Palumbo, Imant Daunhawer, and Julia E Vogt. MMVAE+: Enhancing the generative quality of multimodal VAEs without compromises. In The Eleventh International Conference on Learning Representations, 2023.
>
> [2] Thomas M. Sutter, Imant Daunhawer, and Julia E. Vogt. Generalized multimodal ELBO. In 9th International Conference on Learning Representations, ICLR 2021, Virtual Event, Austria, May 3-7, 2021.
>
> [3] Yusuf Aytar, Carl Vondrick, and Antonio Torralba. Soundnet: Learning sound representations from unlabeled video. Advances in neural information processing systems, 29, 2016.
>
> [4] Xi Chen, Xiao Wang, Soravit Changpinyo, A. J. Piergiovanni, Piotr Padlewski, Daniel Salz, Sebastian Goodman, Adam Grycner, Basil Mustafa, Lucas Beyer, Alexander Kolesnikov, Joan Puigcerver, Nan Ding, Keran Rong, Hassan Akbari, Gaurav Mishra, Linting Xue, Ashish V. Thapliyal, James Bradbury, Weicheng Kuo: PaLI: A Jointly-Scaled Multilingual Language-Image Model. ICLR 2023

---

### Decision · Action_Editor_4nnJ · 2024-12-02

**Recommendation:** Accept with minor revision

**Comment:**

All reviewers recommend acceptance in their final decision. Nevertheless, Reviewer aRPf requests the authors to include the results of baseline video generation models in the additional results of Appendix A.8. The AE acknowledges that such a comparison should be taken carefully, as the different models would be trained with different amounts of data, but this could be highlighted in the table and in the text.

**Audience:**

Yes, all reviewers acknowledge that this work would be of interest to a portion of the TMLR audience.

**Claims And Evidence:**

The reviewers acknowledge that the method is supported by sufficient evidence. Nevertheless, the reviewers also mention that the datasets used in the experiments remain of a fairly small scale.

Furthermore, in their final evaluation, Reviewer dCDR stated that they still "have some doubts about the misalignment between the theoretical conditional independence assumption and their contrastive guidance approach in practice because the assumption itself indicates that no information is shared between different modalities". They nonetheless acknowledged that this work represents "an interesting novel approach that could be interesting for many TMLR audiences working on multi-modal learning".

---

> ### Author Response · Authors · 2024-12-08
>
> Dear reviewers and action editors,
>
> Thank you all for the comments. We have addressed the final comment and uploaded the camera-ready version.